# Efficient Centroid-Linkage Clustering

**MohammadHossein Bateni**
Google Research
New York, USA

**Laxman Dhulipala**
University of Maryland
College Park, USA

**Willem Fletcher**
Brown University
Providence, USA

**Kishen N. Gowda**
University of Maryland
College Park, USA

**D Ellis Hershkowitz**
Brown University
Providence, USA

**Rajesh Jayaram**
Google Research
New York, USA

**Jakub Łącki**
Google Research
New York, USA

## Abstract

We give an algorithm for Centroid-Linkage Hierarchical Agglomerative Clustering (HAC), which computes a $c$-approximate clustering in roughly $n^{1+O(1/c^2)}$ time. We obtain our result by combining a new Centroid-Linkage HAC algorithm with a novel fully dynamic data structure for nearest neighbor search which works under adaptive updates.

We also evaluate our algorithm empirically. By leveraging a state-of-the-art nearest-neighbor search library, we obtain a fast and accurate Centroid-Linkage HAC algorithm. Compared to an existing state-of-the-art exact baseline, our implementation maintains the clustering quality while delivering up to a $36\times$ speedup due to performing fewer distance comparisons.

## 1  Introduction

Hierarchical Agglomerative Clustering (HAC) is a widely-used clustering method, which is available in many standard data science libraries such as SciPy [39], scikit-learn [34], fastcluster [29], Julia [20], R [35], MATLAB [28], Mathematica [25] and many more [31, 32, 36]. HAC takes as input a collection of $n$ points in $\mathbb{R}^d$. Initially, it puts each point in a separate cluster, and then proceeds in up to $n - 1$ steps. In each step, it merges the two "closest" clusters by replacing them with their union.

The formal notion of closeness is given by a *linkage function*; choosing different linkage functions gives different variants of HAC. In this paper, we study *Centroid-Linkage HAC* wherein the distance between two clusters is simply the distance between their centroids. This method is available in most of the aforementioned libraries. Other common choices include *single-linkage* (the distance between two clusters is the *minimum* distance between a point in one cluster and a point in the other cluster) or *average-linkage* (the distance between two clusters is the *average* pairwise distance between them).

HAC's applicability has been hindered by its limited scalability. Specifically, running HAC with any popular linkage function requires essentially quadratic time under standard complexity-theory assumptions. This is because (using most linkage functions) the first step of HAC is equivalent to finding the closest pair among the input points. In high-dimensional Euclidean spaces, this problem was shown to (conditionally) require $\Omega(n^{2-\alpha})$ time for any constant $\alpha > 0$ [23]. In contrast, solving HAC in essentially $\Theta(n^2)$ time is easy for many popular linkage functions (including centroid, single, average, complete, Ward), as it suffices to store all-pairs distances between the clusters in a priority queue and update them after merging each pair of clusters.

38th Conference on Neural Information Processing Systems (NeurIPS 2024).

In this paper, we show how to bypass this hardness for Centroid-Linkage HAC by allowing for approximation. Namely, we give a subquadratic time algorithm which, instead of requiring that the two closest clusters be merged in each step, allows for any two clusters to be merged if their distance is within a factor of $c \geq 1$ of that of the two closest clusters.

**Contribution 1: Meta-Algorithm for Centroid-Linkage.** Our first contribution is a simple meta-algorithm for approximate Centroid-Linkage HAC. The algorithm assumes access to a dynamic data structure for approximate nearest-neighbor search (ANNS). A dynamic ANNS data structure maintains a collection of points $S \subset \mathbb{R}^d$ subject to insertions and deletions and given any query point $u \in S$ returns an approximate nearest neighbor $v \in S$. Formally, for a $c$-approximate NNS data structure, $v$ satisfies $D(u, v) \leq c \cdot \min_{x \in S \setminus \{u\}} D(u, x)$.[1] Here, $D(\cdot, \cdot)$ denotes the Euclidean distance. By using different ANNS data structures we can obtain different Centroid-Linkage HAC algorithms (which is why we call our method a *meta*-algorithm). We make use of an ANNS data structure where the set of points $S$ is the centroids of the current HAC clusters. Roughly, our algorithm runs in time $\tilde{O}(n)$ times the time it takes an ANNS data structure to update or query.

While finding distances is the core part of running HAC, we note that access to an ANNS data structure for the centroids does not immediately solve HAC. Specifically, an ANNS data structure can efficiently find an (approximate) nearest neighbor of a single point, but running HAC requires us to find an (approximately) closest pair among all *clusters*. Furthermore, HAC must use the data structure in a dynamic setting, and so an update to the set of points $S$ (caused by two clusters merging) may result in many points changing their nearest neighbors.

Our meta-algorithm requires that the dynamic ANNS data structure works under *adaptive updates*.[2] Namely, it has to be capable of handling updates, which are dependent on the prior answers that it returned (as opposed to an ANNS data structure which only handles "oblivious" updates).

To illustrate this, consider a randomized ANNS data structure $\mathcal{D}$. Clearly, a query to $\mathcal{D}$ often has multiple correct answers, as $\mathcal{D}$ can return *any* near neighbor within the promised approximation bound. As a result, an answer to a query issued to $\mathcal{D}$ is dependent on the internal randomness of $\mathcal{D}$. Let us assume that $\mathcal{D}$ is used within a centroid-linkage HAC algorithm $\mathcal{A}$ and upon some query returns a result $u$. Then, algorithm $\mathcal{A}$ uses $u$ to decide which two clusters to merge, and the centroid $p$ of the newly constructed cluster is inserted into $\mathcal{D}$. Since $p$ depends on $u$, which in turn is a function of the internal randomness of $\mathcal{D}$, we have that the point that is inserted into $\mathcal{D}$ is dependent on the internal randomness of the data structure. Hence, it is not sufficient for $\mathcal{A}$ to return a correct answer for each *fixed* query with high probability (meaning at least $1 - 1/\text{poly}(n)$). Instead, $\mathcal{A}$ must be able to handle queries and updates dependent on its internal randomness. We note that a similar issue is prevalent in many cases when a randomized data structure is used as a building block of an algorithm. As a result, the subtle notion of adaptive updates is a major area of study [40, 7, 33, 22, 21].

**Contribution 2: Dynamic ANNS Robust to Adaptive Updates.** Our second contribution is a dynamic ANNS data structure for high-dimensional Euclidean spaces which, to the best of our knowledge, is the first one to work under *adaptive updates*.

To obtain our ANNS data structure, we show a black-box reduction from an arbitrary randomized dynamic ANNS data structure to one that works against adaptive updates (see Theorem 2). The reduction increases the query and update times by only a $O(\log n)$ factor. We apply this reduction to a (previously known) dynamic ANNS data structure which is based on locality-sensitive hashing [3, 19] and requires non-adaptive updates.

By combining our dynamic ANNS data structure for adaptive updates with our meta-algorithm, we obtain an $O(c)$-approximate Centroid-Linkage HAC algorithm which runs in time roughly $n^{1+1/c^2}$.

Furthermore, our data structure for dynamic ANNS will likely have applications beyond our HAC algorithm. Specifically, ANNS is commonly used as a subroutine to speed up algorithms for geometric problems. Oftentimes, and similarly to HAC, formal correctness of these algorithms require an ANNS data structure that works for adaptive updates but prior work often overlooks this issue. Our new data structure can be used to fix the analysis of such works. For instance, an earlier work on subquadratic HAC algorithms for average- and Ward-linkage [1] overlooked this issue and our new data structure

---

[1]We note that the notion of ANNS that we use is, in fact, slightly more general than this.
[2]Other papers sometimes refer to this property as working against an *adaptive adversary*.

fixes the analysis of this work.[3] Similarly, a key bottleneck in the dynamic $k$-centers algorithm in [5] was that it had to run nearest neighbor search over the entire dataset (instead of just the $k$-centers). By replacing the ANNS data structure used therein with our ANNS algorithm, we believe that that the $n^\epsilon$ running times from that paper could be improved to $k^\epsilon$.

**Contribution 3: Empirical Evaluation of Centroid-Linkage HAC.** Finally, we use our Centroid-Linkage meta algorithm to obtain an efficient HAC implementation. We use a state-of-the-art static library for ANNS that we extend to support efficient dynamic updates when merging centroids.

We empirically evaluate our algorithm and find that our approximate algorithm achieves strong fidelity with respect to the exact centroid algorithm, obtaining ARI and NMI scores that are within 7% of that of the exact algorithm, while achieving up to $36\times$ speedup over the exact implementation, even when both implementations are run sequentially. Our implementation can be found at `https://github.com/kishen19/CentroidHAC`.

We note that the ANNS we use in the empirical evaluation of our meta-algorithm is different from our new dynamic ANNS (Contribution 2). This is because modern practical methods for ANNS (e.g., based on graph-based indices [38, 18]) have far surpassed the efficiency of data structures for which rigorous theoretical bounds are known. Closing this gap is a major open problem.

## 1.1 Related Work

Improving the efficiency of HAC has been an actively researched problem for over 40 years [8, 31, 30, 32]. A major challenge common to multiple linkage functions has been to improve the running time beyond $\Theta(n^2)$, which is the time it takes to compute all-pairs distances between the input points. For the case of low-dimensional Euclidean spaces, a running time of $o(n^2)$ is possible for the case of squared Euclidean distance and Ward linkage, since the ANNS problem becomes much easier in this case [41]. However, breaking the $\Theta(n^2)$ barrier is (conditionally) impossible in the case of high-dimensional Euclidean spaces (without an exponential dependence on the dimension) [23].

To bypass this hardness result, a recent line of work focused on obtaining *approximate* HAC algorithms [26]. Most importantly, Abboud, Cohen-Addad and Houdrouge [1] showed an approximate HAC algorithm which runs in subquadratic time for Ward and average linkage. These algorithms rely on the fact in the case of these linkage functions the minimum distance between two clusters can only increase as the algorithm progresses, which is not the case for centroid linkage; e.g. consider 3 equi-distant points as in Figure 1. We also note that, as mentioned earlier, our dynamic ANNS

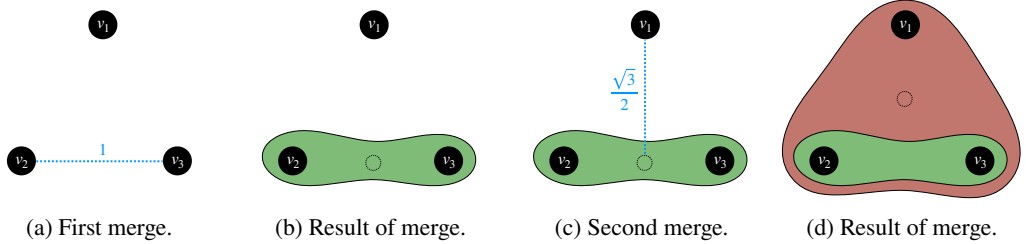

|  |  |  |  |
|---|---|---|---|
| (a) First merge. | (b) Result of merge. | (c) Second merge. | (d) Result of merge. |

Figure 1: 3 points in $\mathbb{R}^2$ initially all at distance 1 showing Centroid-Linkage HAC merge distances can decrease. 1a / 1b and 1c / 1d give the the first and second merges with distances 1 and $\sqrt{3}/2 < 1$ respectively.

data structure fixes a gap in the analysis of both algorithms in the paper. In fact, not only can the minimum distances shrink when performing Centroid-Linkage HAC, but when doing $O(1)$-approximate Centroid-Linkage HAC distances can get arbitrarily small; see Figure 2. This presents an additional issue for dynamic ANNS data structures which typically assume lower bounds on minimum distances.

Compared to the algorithm of Abboud, Cohen-Addad and Houdrouge, our centroid-linkage HAC algorithm introduces two new ideas. First, we show how to handle the case when the minimum

---

[3]We contacted the authors of [1] and they confirmed this gap in their analysis.

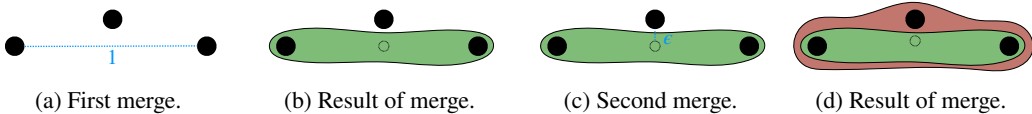

(a) First merge.      (b) Result of merge.      (c) Second merge.      (d) Result of merge.

Figure 2: 3 points in $\mathbb{R}^2$ (two of which are initially at distance 1) showing that $O(1)$-approximate Centroid-Linkage HAC can arbitrarily reduce merge distances. 2a / 2b and 2c / 2d give the the first and second merges with distances 1 and $\epsilon \ll 1$ respectively; centroids are dashed circles.

distance between two clusters decreases as a result of two cluster merging. Second, we introduce an optimization that allows us not to consider each cluster at each of the (logarithmically many) distance scales. This optimization improves the running time bound by a logarithmic factor when each merge "makes stale" a small number of stored pairs and in practice the number of such stale merges occur only $60\%$ of the time (on average) compared to the number of actual merges performed.

Another line of work considered a different variant of HAC, where the input is a weighted similarity graph [15–17, 6]. The edge weights specify similarities between input points, and a lack of edge corresponds to a similarity value of 0. By assuming that the graph is sparse, this representation allows bypassing the hardness of finding distances, which leads to near-linear-time approximate algorithms for average-linkage HAC [15], as well as to efficient parallel algorithms [16, 6].

In terms of the theoretical and empirical part of this paper considering different algorithms, such a gap has also been observed for various other prominent clustering algorithms. For instance, for correlation clustering, while the best known algorithms are obtained by solving a linear program (see [9]), in practice a simple local search/Louvain-based algorithm is used (see section 4.1 of [2]). For modularity clustering, the best known approximation is obtained by solving an SDP [24]. Again, Louvain-based algorithms are used in practice. In the case of k-means, while the best known approximation is a constant obtained via the primal-dual method [10], in practice Lloyd's heuristic paired with k-means++ seeding is used which has a logarithmic approximation guarantee. For balanced graph partitioning, coarsening combined with a brute-force algorithm is used to obtain the best results in practice. On the other hand, the algorithms for solving the corresponding theoretical formulations, i.e. balanced cut and multiway cut, are entirely different.

## 2 Preliminaries

In this section, we review our conventions and preliminaries. Throughout this paper, we will work in $d$-dimensional Euclidean space $\mathbb{R}^d$. We will use $D(\cdot, \cdot)$ for the Euclidean metric in $\mathbb{R}^d$; that is, for $x, y \in \mathbb{R}^d$, the distance between $x$ and $y$ is $D(x, y) := \sqrt{\sum_i (x[i] - y[i])^2}$ where $x[i]$ and $y[i]$ are the $i$th coordinates of $x$ and $y$ respectively. Likewise, we let $D(x, Y) := \min_{y \in Y} D(x, y)$ for $Y \subseteq \mathbb{R}^d$.

### 2.1 Formal Description of Centroid-Linkage HAC

While we have described centroid-linkage in a cluster-centric way (see Appendix A for a formal cluster-centric definition), it will be more useful to use an equivalent centroid-centric definition.

Specifically, we can equivalently define $c$-approximate Centroid-Linkage HAC as repeatedly "merging centroids". Initialize the set of all centroids $C = P$ and a weight $w_u = 1$ for each $u \in C$; these weights will encode the cluster sizes of each centroid. Then, until $|C| = 1$ we do the following: let $\hat{x}, \hat{y} \in C$ be a pair of centroids satisfying

$$D(\hat{x}, \hat{y}) \leq c \cdot \min_{x, y \in C} D(x, y)$$

where the $\min$ is taken over distinct pairs; merge $\hat{x}$ and $\hat{y}$ into their weighted midpoints by removing them from $C$, adding $z = (w_{\hat{x}}\hat{x} + w_{\hat{y}}\hat{y})/(w_{\hat{x}} + w_{\hat{y}})$ to $C$ and setting $w_z = w_{\hat{x}} + w_{\hat{y}}$. Also, note that (exact) Centroid-Linkage HAC is just the above with $c = 1$.

### 2.2 Dynamic Nearest-Neighbor Search

We define a dynamic ANNS data structure. Typically, dynamic ANNS requires a lower bound on distances; we cannot guarantee this for centroid HAC. Thus, we make use of $\beta$ additive error below.

**Definition 1** (Dynamic Approximate Nearest-Neighbor Search (ANNS)). *An $(\alpha, \beta)$-approximate dynamic nearest-neighbor search data structure $\mathcal{N}$ maintains a dynamically updated set $S \subseteq \mathbb{R}^d$ (initially $S = \emptyset$) and supports the following operations.*

1. ***Insert**, $\mathcal{N}$.INSERT(u). given $u \notin S$ update $S \leftarrow S \cup \{u\}$.*
2. ***Delete**, $\mathcal{N}$.DELETE(u): given $u \in S$ update $S \leftarrow S \setminus \{u\}$;*
3. ***Query**, $\mathcal{N}$.QUERY($u, \bar{u}$): given $u, \bar{u} \in \mathbb{R}^d$ return $v \in S \setminus \{\bar{u}\}$ s.t.*

$$D(u, v) \le \alpha \cdot D(u, S \setminus \{\bar{u}\}) + \beta.$$

If $\mathcal{N}$ is a dynamic $(\alpha, 0)$-approximate NNS data structure then we will simply call it $\alpha$-approximate. For a set of points $S$, we will use the notation NonAdaptiveAnn($S$) to denote the result of starting with an empty dynamic ANNS data structure and inserting each point in $S$ (in an arbitrary order).

We say $\mathcal{N}$ *succeeds for a set of points $S \subseteq \mathbb{R}^d$ and a query point $u$* if after starting with an empty set and any sequence of insertions and deletions of points in $S$ we have that the query $\mathcal{N}$.QUERY($u, \bar{u}$) is correct for any $\bar{u}$ (i.e., the returned point satisfies the stated condition in 3). If the data structure is randomized then we say that it *succeeds for adaptive updates* if with high probability (over the randomness of the data structure) it succeeds for all possible subsets of points and queries. Queries have *true distances* at most $\Delta$ if the (true) distance from $S$ to any query is always at most $\Delta$.

The starting point for our dynamic ANNS data structure is the following data structure requiring *non-adaptive* updates which alone does not suffice for centroid HAC as earlier described. See Appendix B for a proof.

**Theorem 1** (Dynamic ANNS for Oblivious Updates, [3, 19]). *Suppose we are given $\gamma > 1$ and $c, \beta, \Delta$ and $n$ where $\log(\Delta/\beta), \gamma \le \mathsf{poly}(n)$ and a dynamically updated set $S$ with at most $n$ insertions. Then, if all queries have true distance at most $\Delta$, we can compute a randomized $(O(c), \beta)$-approximate NNS data structure with update, deletion and query times of $n^{1/c^2 + o(1)} \cdot \log(\Delta/\beta) \cdot d \cdot \gamma$, which for a fixed set of points and query point succeeds except with probability at most $\exp(-\gamma)$.*

## 3 Dynamic ANNS with Adaptive Updates

The main theorem we show in this section is how to construct a dynamic ANNS data structure that succeeds for adaptive updates using one which succeeds for oblivious updates.

**Theorem 2** (Reduction of Dynamic ANNS from Oblivious to Adaptive). *Suppose we are given $c$, $\beta$, $\Delta$, $n$, $s \in \mathbb{R}^d$ and dynamically updated set $S \subset \mathbb{R}^d$ with at most $n$ insertions, such that all inserted points and query points lie in $B_s(\Delta)$. Moreover, assume that we can compute a dynamic $(c, \beta)$-approximate NNS data structure with query, deletion and insertion times $T_Q$, $T_D$ and $T_I$ which succeeds for $S$ and a fixed query except with probability at most $n^{-O(d \log d \log(\Delta/\beta))}$.*

*Then, we can compute a randomized dynamic $(c, c\beta)$-approximate NNS data structure that succeeds for adaptive updates with query, deletion and amortized insertion times $O(\log n) \cdot T_Q$, $O(T_D)$ and $O(\log n) \cdot T_I$.*

We note that our result is slightly stronger than the above: the insertions we make into the oblivious ANNS that we use only occur upon their instantiation, not dynamically.

As a corollary of the above reduction and known constructions for dynamic ANNS that work for oblivious updates—namely, Theorem 1 with $\gamma = \Theta(d \log d \cdot \log(\Delta/\beta) \cdot \log n)$ and using parameter $\beta' = \beta/c$ where $\beta'$ is the additive distortion parameter for Theorem 1—we obtain the following.

**Theorem 3** (Dynamic ANNS for Adaptive Updates). *Suppose we are given $c$, $\beta$, $\Delta$, $n$, $s \in \mathbb{R}^d$ where $\log(\Delta/\beta), d, c \le \mathsf{poly}(n)$ and dynamically updated set $S \subset \mathbb{R}^d$ with at most $n$ insertions, such that all inserted and query points lie in $B_s(\Delta)$.*

*Then, we can compute a randomized dynamic $(O(c), \beta)$-approximate NNS data structure that succeeds for adaptive updates with query, deletion and amortized insertion time $n^{1/c^2 + o(1)} \cdot \log(\Delta/\beta) \cdot d^2$.*

A previous work [19] also provided algorithms that work against an "adaptive updates" but only a set of adaptive updates made against a fixed set of query points. Also, note that if we have a lower bound of $\delta$ on the true distance of any query then lowering $c$ by a constant and setting $\beta = \Theta(\delta)$ for a suitably small hidden constant gives an $O(c)$-approximate NNS with similar guarantees.

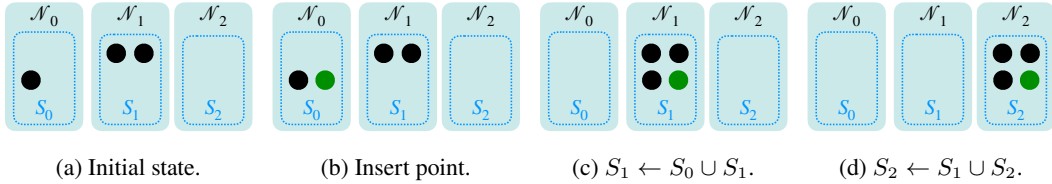

| (a) Initial state. | (b) Insert point. | (c) $S_1 \leftarrow S_0 \cup S_1$. | (d) $S_2 \leftarrow S_1 \cup S_2$. |

Figure 3: Our merge-and-reduce strategy when a point (in green) is inserted.

## 3.1 Algorithm Description

Our algorithm for dynamic ANNS for adaptive updates uses two ingredients. First, we make use of the following "covering nets" to fix our queries to a small set against which we can union bound.

**Lemma 1.** *Given $s \in \mathbb{R}^d$ and $\beta, \Delta > 0$, there exists a* covering net $Q \subseteq \mathbb{R}^d$ *such that*

1. **Small Size:** $|Q| \leq (\Delta/\beta)^{O(d \log d)}$
2. **Queries:** *given $u \in B_s(\Delta)$, one can compute $u' \in Q$ in time $O(d)$ such that $D(u, u') \leq \beta$.*

Second, we make use of a "merge-and-reduce" approach. A similar approach was taken by [12] for exact $k$-nearest-neighbor queries in the plane. Namely, for each $i \in [O(\log n)]$ we maintain a set $S_i$ of size at most $2^i$ (where all $S_i$ partition all inserted points) and a dynamic ANNS data structure $\mathcal{N}_i$ for $S_i$ which only works for oblivious updates. Other than the size constraint, the partition is arbitrary. Informally, we perform deletions, insertions and queries as follows.

**Deletion:** To delete a point we simply delete it from its corresponding $\mathcal{N}_i$.

**Insertion:** To insert a point, we insert it into $S_0$ and for each $i$ we move all points from $S_i$ to $S_{i+1}$ if $S_i$ contains more than $2^i$ points; we update $\mathcal{N}_i$ accordingly each time we move points; namely, we recompute $\mathcal{N}_i$ and $\mathcal{N}_{i+1}$ from scratch on $S_i$ and $S_{i+1}$ respectively. See Figure 3.

**Query:** Lastly, to query a point $u$ we first map this point to a point in our covering net $u'$ (as specified by Lemma 1), query $u'$ in each of our $\mathcal{N}_i$ and then return the best output (i.e., the point output by an $\mathcal{N}_i$ that is closest to $u'$).

Our algorithm is more formally described in pseudo-code in Algorithm 1.

---

**Algorithm 1** Dynamic ANNS for Adaptive Updates

---

**Input:** $\beta, \Delta, n, s \in \mathbb{R}^d, Q \leftarrow \text{CoveringNet}(s, \Delta, \beta)$ (computed using Lemma 1)
**Maintains:** $S_0, S_1, \ldots$ – partition of the inserted points, s.t. $|S_i| \leq 2^i$
**Maintains:** $\mathcal{N}_0, \mathcal{N}_1, \ldots$ – a non-adaptive ANNS for each $S_i$
**function** INSERT($u$)
    $S_0 \leftarrow S_0 \cup \{u\}$
    **while** $\exists i$ such that $|S_i| > 2^i$ **do**
        $S_{i+1} \leftarrow S_{i+1} \cup S_i$ and $S_i \leftarrow \emptyset$
        $\mathcal{N}_{i+1} \leftarrow \text{NonAdaptiveAnn}(S_{i+1})$ and $\mathcal{N}_i \leftarrow \text{NonAdaptiveAnn}(\emptyset)$
**function** DELETE($u$)
    Let $i$ be such that $u \in S_i$
    $\mathcal{N}_i$.DELETE($u$) and remove $u$ from $S_i$
**function** QUERY($u, \bar{u}$)
    Let $u' \in Q$ be such that $D(u, u') = O(\beta)$                      $\triangleright$ computed using Lemma 1
    Let $v_i = \mathcal{N}_i$.QUERY($u', \bar{u}$)
    **return** $v = \arg\min_{v_i} D(u', v_i)$

---

# 4 Centroid-Linkage HAC Algorithm

In this section, we give our algorithm for Centroid-Linkage HAC. Specifically, we show how to (with Algorithm 3) reduce approximate Centroid-Linkage HAC to roughly linearly-many dynamic ANNS function calls, provided the ANNS works for adaptive updates.

**Theorem 4** (Reduction of Centroid HAC to Dynamic ANN for Adaptive Updates). *Suppose we are given a set of $n$ points $P \subset \mathbb{R}^d$, $c > 1$, $\epsilon > 0$, and lower and upper bounds on the minimum and maximum pairwise distance in $P$ of $\delta$ and $\Delta$ respectively. Suppose we are also given a data structure that is a dynamic $c$-approximate NNS data structure for all queries in Algorithm 3 that works for adaptive updates with insertion, deletion and query times of $T_I$, $T_D$, and $T_Q$.*

*Then there exists an algorithm for $c(1 + \epsilon)$-approximate Centroid-Linkage HAC on $n$ points that runs in time $\tilde{O}(n \cdot (T_I + T_D + T_Q))$ assuming $\frac{\Delta}{\delta} \leq \mathsf{poly}(n)$ and $\frac{1}{\epsilon} \leq \mathsf{polylog}(n)$.*

Our final algorithm for centroid HAC is not quite immediate from the above reduction and our dynamic ANNS algorithm for adaptive updates. Specifically, the above reduction requires a $c$-approximate NNS data structure but in the preceding section we have only provided a $(c, \beta)$-approximate NNS data structure for $\beta > 0$. However, by leveraging the structure of centroid HAC—in particular, the fact that there is only ever at most one "very close" pair—we are able to show that this suffices. In particular, for a given $c$, there exists a $c_0$ and $\beta_0$ such that a $(c_0, \beta_0)$-approximate NNS data structure functions as a $c$-approximate NNS data structure for all queries in Algorithm 3. This is formalized by Lemma 6. Combining Lemma 6, Theorem 4 and Theorem 3, we get the following.

**Theorem 5** (Centroid HAC Algorithm). *For $n$ points in $\mathbb{R}^d$, there exists a randomized algorithm that given any $c > 1.01$ and lower and upper bounds on the minimum and maximum pairwise distance in $P$ of $\delta$ and $\Delta$ respectively, computes a $c$-approximate Centroid-Linkage HAC with high probability. It has runtime $O(n^{1+O(1/c^2)}d^2)$ assuming $c, d, \frac{\Delta}{\delta} \leq \mathsf{poly}(n)$.*

## 4.1 Algorithm Description

In what remains of this section, we describe the algorithm for Theorem 4. Consider a set of points $P \in \mathbb{R}^d$. Suppose we have a dynamic $c$-approximate NNS data structure $\mathcal{N}$ that works with adaptive updates with query time $T_Q$, insertion time $T_I$, and deletion time $T_D$. Let $Q$ be a priority queue storing elements of the form $(l, x, y)$ where $x$ and $y$ are "nearby" centroids and $l = D(x, y)$. The priority queue supports queuing elements and dequeueing the element with shortest distance, $l$. At any given time while running the algorithm, say a centroid $x$ is queued if there is an element of the form $(l, x, y)$ in $Q$. We will maintain a set, $C$, of active centroids allowing us to check if a centroid is active in constant time. For each active centroid, $C$ will also store the weight of the centroid so we can preform merges and an identifier to distinguish between distinct clusters with the same centroid. Note that any time we store a centroid, including in $Q$, we will implicitly store this identifier.

First, we describe how the algorithm handles merges. To merge two active centroids, $x$ and $y$, we remove them from $C$ and delete them in $\mathcal{N}$. Let $z = w_x x + w_y y$ be the centroid formed by merging $x$ and $y$. We use $\mathcal{N}$ to find an approximate nearest neighbor, $y^*$, of $z$ and add to $Q$ the tuple $(D(z, y^*), z, y^*)$. Lastly, we add $z$ to $\mathcal{N}$ and $C$. Pseudo-code for this algorithm is given by Algorithm 2. Crucially, we do not try to update any nearest neighbors of any other centroid at this stage. We will do this work later and only if necessary. Since we are using an approximate NNS, it is possible that the centroid $z$ will be the same point in $\mathbb{R}^d$ as the centroid of another cluster. We can detect this in constant time using $C$ and we will immediately merge the identical centroids.

We now describe the full algorithm using the above merge algorithm; we also give pseudo-code in Algorithm 3. Let $\epsilon > 0$ be a parameter that tells the algorithm how aggressively to merge centroids. To begin, we construct $\mathcal{N}$ by inserting all points of $P$ in any order. Then for each $p \in P$, we use $\mathcal{N}$ to find an approximate nearest neighbor $y \in P \setminus \{p\}$ and queue $(D(p, y), p, y)$ to $Q$. We also initialize $C = \{p : p \in P\}$. The rest of the algorithm is a while loop that runs until $Q$ is empty. Each iteration of the while loop begins by dequeuing from $Q$ the tuple $(l, x, y)$ with minimum $l$.

**(1)** If $x$ and $y$ are both active centroids then we merge them.
**(2)** Else if $x$ is not active then we do nothing and move on to the next iteration of the while loop.
**(3)** Else if $x$ is active but $y$ is not, we use $\mathcal{N}$ to compute a new approximate nearest neighbor, $y^*$, of $x$. Let $l^* = D(x, y^*)$. If $l^* \leq (1 + \epsilon)l$, then we merge $x$ and $y^*$. Otherwise we add $(l^*, x, y^*)$ to $Q$.

**Algorithm 2** Handle-Merge

1: **Input:** set of active centroids $C$, ANNS data structure $\mathcal{N}$, priority queue $Q$, centroids $x$ and $y$
2: $z \leftarrow w_x x + w_y y$                                                            ▷ Merge $x$ and $y$
3: Remove $x$ and $y$ from $C$ and $\mathcal{N}$
4: **if** there is a centroid $z^*$ that is the same as $z$ is in $C$ **then**
5:     $z \leftarrow w_z z + w z^* z^*$                                         ▷ Merge $z$ and $z^*$
6:     Remove $z^*$ from C and add $z$
7: **else**
8:     Add $z$ to $\mathcal{N}$ and $C$
9: $y^* \leftarrow$ An approximate nearest neighbor of $z$ returned by $\mathcal{N}$.QUERY$(z, z)$
10: Queue $(D(z, y^*), z, y^*)$ to $Q$

---

**Algorithm 3** Approximate-HAC

1: **Input:** set of points $P$, metric $D$, fully dynamic ANNS $\mathcal{N}$ data structure, $\epsilon > 0$
2: **Output:** $(1 + \epsilon)c$-approximate centroid HAC
3: Initialize $Q$ and $C = \{\{p\} : p \in P\}$ and insert all points of P into $\mathcal{N}$
4: **for** $p \in P$ **do**
5:     $y \leftarrow$ An approximate nearest neighbor of $p$ returned by $\mathcal{N}$.QUERY$(p, p)$
6:     Queue $(D(p, y), p, y)$ to $Q$
7: **while** $Q$ is not empty **do**
8:     $(l, x, y) \leftarrow$ dequeue shortest distance from $Q$
9:     **if** $x, y \in C$ **then**
10:         Handle-Merge$(C, \mathcal{N}, Q, x, y)$
11:     **else if** $x \in C$ **then**
12:         $y^* \leftarrow$ An approximate nearest neighbor of $x$ returned by $\mathcal{N}$.QUERY$(x, x)$
13:         $l^* \leftarrow D(x, y^*)$
14:         **if** $l^* \leq (1 + \epsilon)l$ **then**
15:             Handle-Merge$(C, \mathcal{N}, Q, x, y^*)$
16:         **else**
17:             Queue $(l^*, x, y^*)$ to $Q$

## 5 Empirical Evaluation

We empirically evaluate the performance and effectiveness of our approximate Centroid HAC algorithm through a comprehensive set of experiments. Our analysis on various publicly-available benchmark clustering datasets demonstrates that our approximate Centroid HAC algorithm:

(1) Consistently produces clusters with quality comparable to that of exact Centroid HAC,
(2) Achieves an average speed-up of $22\times$ with a max speed-up of $175\times$ compared to exact Centroid HAC when using 96 cores in our scaling study. On a single core, it achieves an average speed-up of $5\times$ with a max speed-up of $36\times$.

**Dynamic ANNS Implementation using DiskANN** As demonstrated in previous works, although LSH-based algorithms have strong theoretical guarantees, they tend to perform poorly in practice when compared to graph-based ANNS data structures [18]. For instance, in ParlayANN [18], the authors find that the recall achievable by LSH-based methods on a standard ANNS benchmark dataset are strictly dominated by state-of-the-art graph-based ANNS data structures.

Therefore, for our experiments, we use DiskANN, a widely-used state-of-the-art graph-based ANNS data structure [38]. In particular, we consider the in-memory version of this algorithm from ParlayANN, called Vamana. In a nutshell, the algorithm builds a bounded degree *routing* graph that can be searched using beam search. Note that this graph is *not* the $k$-NN graph of the pointset. The graph is constructed using an incremental construction that adds bidirectional edges between a newly inserted point, and points traversed during a beam search for this point; if a point's degree exceeds the degree bound, the point is *pruned* to ensure a diverse set of neighbors. See [38] for details.

For Centroid HAC, we require the ANNS implementation to support dynamic updates. The implementation provided by ParlayANN currently only supports fast *static* index building and queries. Recently, FreshDiskANN [37] described a way to handle insertions and deletions via a *lazy update*

Table 1: The ARI and NMI scores of our approximate Centroid HAC implementations for $\epsilon = 0.1, 0.2, 0.4$, and $0.8$, versus Exact Centroid HAC. The best quality score for each dataset is in bold and underlined.

| | Dataset | Centroid$_{0.1}$ | Centroid$_{0.2}$ | Centroid$_{0.4}$ | Centroid$_{0.8}$ | Exact Centroid |
|---|---|---|---|---|---|---|
| ARI | iris | **0.759** | 0.746 | 0.638 | 0.594 | **0.759** |
| | wine | 0.352 | 0.352 | **0.402** | 0.366 | 0.352 |
| | cancer | 0.509 | 0.526 | 0.490 | **0.641** | 0.509 |
| | digits | 0.589 | 0.571 | 0.576 | **0.627** | 0.559 |
| | faces | 0.370 | 0.388 | **0.395** | 0.392 | 0.359 |
| | mnist | **0.270** | 0.222 | 0.218 | 0.191 | 0.192 |
| | birds | 0.449 | 0.449 | 0.442 | **0.456** | 0.441 |
| | **Avg** | **0.471** | 0.465 | 0.452 | 0.467 | 0.453 |
| NMI | iris | **0.803** | 0.795 | 0.732 | 0.732 | **0.803** |
| | wine | 0.424 | 0.424 | 0.413 | 0.389 | 0.424 |
| | cancer | 0.425 | 0.471 | 0.459 | **0.528** | 0.425 |
| | digits | 0.718 | 0.726 | 0.707 | **0.754** | 0.727 |
| | faces | 0.539 | 0.534 | 0.549 | 0.549 | **0.556** |
| | mnist | 0.291 | 0.282 | 0.306 | **0.307** | 0.250 |
| | birds | 0.748 | 0.747 | 0.756 | **0.764** | 0.743 |
| | **Avg** | 0.564 | 0.569 | 0.560 | **0.575** | 0.561 |

approach. However, their approach requires a periodic *consolidation* step that scans through the graph and deletes inactive nodes and rebuilds the neighborhood of affected nodes, which is expensive.

Instead, in our implementation of Centroid HAC, we adopt the following new approach that is simple and practical, and adheres to the updates required by our theoretical algorithm: when clusters $u$ and $v$ merge, choose one of them to represent the centroid, and update its neighborhood $N(u)$ (without loss of generality) to the set obtained by pruning the set $N(u) \cup N(v)$. The intuition here is that $N(u) \cup N(v)$ is a good representative for the neighborhood of the centroid of $u$ and $v$ in the routing graph. Further, during search, points will redirect to their current representative centroid (via union-find [14]), thus allowing us to avoid updating the in-neighbors of a point in the index. We believe application-driven update algorithms, such as the one described here, can help speed-up algorithms that uses dynamic ANNS as a subroutine.

## 5.1 Quality Evaluation

We evaluate the clustering quality of our approximate centroid HAC algorithm against ground truth clusterings using standard metrics such as the *Adjusted Rand Index (ARI)*, *Normalized Mutual Information (NMI)*, *Dendrogram Purity*, and the unsupervised *Dasgupta cost* [11]. Our primary objective is to assess the performance of our algorithm across various values of $\epsilon$ on a diverse set of benchmarks. We primarily compare our results with those obtained using exact Centroid HAC.

For these experiments, we consider the standard benchmark clustering datasets `iris`, `digits`, `cancer`, `wine`, and `faces` from the UCI repository (obtained from the `sklearn.datasets` package). We also consider the `MNIST` dataset which contains images of grayscale digits between 0 and 9, and `birds`, a dataset containing images of 525 species of birds; see Appendix E for more details.

**Results.** The experimental results are presented in Table 1, with a more detailed quality evaluation in Appendix E.1. We summarize our results here.

We observe that the quality of the clustering produced by approximate Centroid HAC is generally comparable to that of the exact Centroid HAC algorithm. On average, the quality is slightly better in some cases, but we attribute this to noise. In particular, we observe that for the value of $\epsilon = 0.1$, we consistently get comparable quality to that of exact centroid: the ARI and NMI scores are on average within a factor of 7% and 2% to that of exact Centroid HAC, respectively. We also obtained good dendrogram purity score and Dasgupta cost as well, with values within 0.3% and 0.03%, respectively.

## 5.2 Running Time Evaluation

Next, we evaluate the scalability of approximate Centroid HAC against exact Centroid HAC on large real-world pointsets. We consider the optimized exact Centroid HAC implementation from the `fastcluster` package [29]. We also implement an efficient version of exact Centroid HAC based on our framework (i.e. setting $\epsilon = 0$ and using exact NNS queries) which has the benefit of using only linear space and supports parallelism in NNS queries. We also implement a *bucket-based* version of approximate Centroid HAC as a baseline, based on the approach of [1] along with an observation to handle the non-monotonicity of Centroid HAC; details in Appendix E.

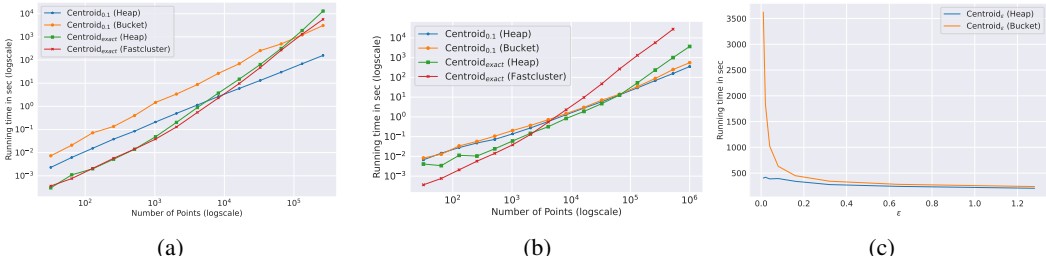

Figure 4: Running times of fastcluster's centroid HAC, our implementation of exact centroid HAC, and the heap-based and bucket-based approximate centroid HAC with $\epsilon = 0.1$. In Figure 4a, our approximate and exact implementations are run on 1 core, whereas in Figure 4b they have access to 192 cores. Figure 4c compares the running times of heap and bucket based algorithms as a function of $\epsilon$.

**Results.** We now summarize the results of our scalability study; see Appendix E.2 for a more details and plots. Figures 4a and 4b shows the running times on varying slices of the `SIFT-1M` dataset using one thread and 192 parallel threads, respectively. For the approximate methods, we considered $\epsilon = 0.1$. Figure 4c further compares the heap and bucket based approaches as a function of $\epsilon$.

We observe that the bucket based approach is very slow for small values of $\epsilon$ due to many redundant nearest-neighbor computations. Yet, at $\epsilon = 0.1$, both algorithms, with 192 threads, obtain speed-ups of up to $175\times$ with an average speed-up of $22\times$ compared to the exact counterparts. However, on a single thread, the bucket based approach fails to scale, while the heap based approach still achieves significant speed-ups of upto $36\times$ with an average speed-up of $5\times$. Overall, our heap-based approximate Centroid HAC implementation with $\epsilon = 0.1$ demonstrates good quality and scalability, making it a good choice in practice.

## 6 Conclusion

In this work we gave an approximate algorithm for Centroid-Linkage HAC which runs in subquadratic time. Our algorithm is obtained by way of a new ANNS data structure which works correctly under adaptive updates which may be of independent interest. On the empirical side we have demonstrated up to $36\times$ speedup compared to the existing baselines. An interesting open question is whether approximate Centroid-Linkage HAC admits a theoretically and practically efficient *parallel* algorithm.

## Acknowledgments

This work was supported by NSF grants CNS-2317194 and CCF-2403235. We thank Vincent Cohen-Addad for helpful discussions. We thank the anonymous reviewers for their useful comments.

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

## A  Cluster-Centric Definition of Centroid-Linkage HAC

We formally define Centroid-Linkage HAC in a cluster-centric way. For a set of points $X \subset \mathbb{R}^d$, we define $\text{cent}(X)$ to be the centroid of $X$ as below.

**Definition 2** (Centroid). *Given a set of points $X \subset \mathbb{R}^d$, the centroid of $X$ is the point $x \in \mathbb{R}^d$ whose ith coordinate is*

$$x[i] = \sum_{y \in X} \frac{y[i]}{|X|}.$$

Centroid HAC uses distance between centroids as a linkage function. That is, for clusters $X$ and $Y$,

$$\mathcal{L}(X, Y) = D(\text{cent}(X), \text{cent}(Y)).$$

$c$-approximate Centroid-Linkage HAC is defined as follows. We are given input points $P \subset \mathbb{R}^d$. First, initialize a set of active clusters $\mathcal{C} = \{\{p\}\}_{p \in P}$. Then, until $|\mathcal{C}| = 1$ we do the following: let $\hat{X}, \hat{Y} \in \mathcal{C}$ be a pair of active clusters satisfying

$$\mathcal{L}(\hat{X}, \hat{Y}) \leq c \cdot \min_{X,Y \in \mathcal{C}} \mathcal{L}(X, Y)$$

where the $\min$ is taken over distinct pairs; merge $\hat{X}$ and $\hat{Y}$ by removing them from $\mathcal{C}$ and adding $\hat{X} \cup \hat{Y}$ to $\mathcal{C}$. (Non-approximate) Centroid-Linkage HAC is just the above with $c = 1$.

## B  Proof of Theorem 1

In the interest of completion, we give a proof of Theorem 1 (dynamic ANNS for oblivious updates).

### B.1  Locality Sensitive Hashing

We will make use of the well-known algorithmic primitive of locality sensitive hashing.

**Definition 3** (Locality Sensitive Hashing). *A family of hash functions $\mathcal{H}$ is called $(r, cr, p_1, p_2)$-sensitive if for any $p, q \in \mathbb{R}^d$ we have*

1. ***Close Points Probably Collide:*** *If $D(u, v) \leq r$ then $\Pr_{h \sim H}(h(u) = h(v)) \geq p_1$.*

2. ***Far Points Probably Don't Collide:*** *If $D(u, v) \geq cr$ then $\Pr_{h \sim \mathcal{H}}(h(u) = h(v)) \leq p_2$.*

We will make use of known locality sensitive hashing schemes for Euclidean distance in $\mathbb{R}^d$. For this result, see also [4].

**Theorem 6** (Lemma 3.2.3 of [3]). *Given $r$ and $c$, there is a $(r, cr, p_1, p_2)$-sensitive LSH family with*

$$\frac{\log(1/p_1)}{\log(1/p_2)} = 1/c^2 + o(1) \qquad \text{and} \qquad \log(1/p_2) \leq o(\log n).$$

*Furthermore, one can sample $h \sim \mathcal{H}$ and given $u \in \mathbb{R}^d$ compute $h(u)$ in time $dn^{o(1)}$.*

**Theorem 1** (Dynamic ANNS for Oblivious Updates, [3, 19]). *Suppose we are given $\gamma > 1$ and $c, \beta, \Delta$ and $n$ where $\log(\Delta/\beta), \gamma \leq \text{poly}(n)$ and a dynamically updated set $S$ with at most $n$ insertions. Then, if all queries have true distance at most $\Delta$, we can compute a randomized $(O(c), \beta)$-approximate NNS data structure with update, deletion and query times of $n^{1/c^2+o(1)} \cdot \log(\Delta/\beta) \cdot d \cdot \gamma$, which for a fixed set of points and query point succeeds except with probability at most $\exp(-\gamma)$.*

*Proof.* We first construct a data structure which succeeds with constant probability for a given query. To do so we begin by turning our LSH hashes from Theorem 6 into a hash function which is an "$L$ ors of $K$ ands". In particular, we let

$$L := n^{1/c^2+o(1)} \cdot \Theta(\log n).$$

and let

$$K := \frac{1}{\log(1/p_2)} \cdot \left(\Theta(\log n) + \log L + \log\log(\Delta/\beta)\right).$$

Notice that by Theorem 6 we have that $\log(1/p_2) \leq o(\log n)$ and by assumption we have $\log(\Delta/\beta), \gamma \leq \text{poly}(n)$ and so we have

$$K \cdot L \leq n^{1/c^2 + o(1)} \tag{1}$$

For each $i$ such that $\beta \leq 2^i \leq \Delta$, we define $L$-many new hash functions, each of which consists of $K$-many $h$ from Theorem 6. In particular, for $i \in [\log(\Delta/\beta)]$, $l \in [L]$ and $k \in [K]$, we let $h_{ikl}$ be a uniformly random $h$ sampled according to Theorem 6 using radius $r = 2^i$ and our input $c$. Likewise, we let $g_{il} : \mathbb{R}^d \to \mathbb{R}^k$ be a new hash function defined on $u \in \mathbb{R}^d$ as

$$g_{il}(u) = (h_{il1}(u), h_{il2}(u), \ldots, h_{ilk}(u)).$$

We let $G_i := \{g_{il} : l \leq L\}$ be all such hash functions for $i$ and let $\mathcal{G} = \bigcup_i G_i$ be all such hash functions across scales.

By Theorem 6, sampling each constituent hash functions for each $g \in \mathcal{G}$ can be done in time $dn^{o(1)}$ and so by this and Equation (1) initializing all hash functions in $\mathcal{G}$ takes time

$$dn^{o(1)} \cdot L \cdot K \cdot \log(\Delta/\beta) = n^{1/c^2 + o(1)} \cdot \log(\Delta/\beta) \cdot d \tag{2}$$

This initialization time will be dominated by the time it takes to insert a single point. For each such hash function we will maintain a collection of buckets which partitions points in $S$.

To insert a point into $S$, we simply compute its hash for each $g \in \mathcal{G}$ and store this hash in each corresponding bucket. To delete a point from $S$ we simply delete it from all buckets to which it hashes. Lastly, to answer a query on $(u, \bar{u})$, we hash $u$ according to each $g \in \mathcal{G}$ and then return the $v \in S \setminus \bar{u}$ such that for minimum $i$ we have a $g \in G_i$ such that $g(u) = g(v)$. Since by Theorem 6 evaluating each hash function takes time $dn^{o(1)}$, we get that all of these operations also take time at most that given by Equation (2).

Lastly, we argue the (constant probability) correctness of a query for a fixed set of points. Fix query points $(u, \bar{u})$ and a set of points $S$. Let $w \in S \setminus \{u\}$ be a fixed point of $S \setminus \{u\}$ where we let $i_w$ be such that $2^{i_w} \leq D(u, w) \leq 2^{i_w + 1}$. Say that $w$ *succeeds* if

1. **One Correct Collision:** there is a $g \in G_{i_w}$ such that $g(w) = g(u)$ and

2. **No Incorrect Collisions:** for all $i < i_w$ and $g \in G_i$ we have that $g(u) \neq g(w)$.

Observe that if both of the cases happen for every $w \in S \setminus \{u\}$ then we have that our query is correct. The only non-trivial part of this observation is the fact that if $u$ is within distance $\beta$ of $S \setminus \{\bar{u}\}$ then $i_w = 1$ and the returned point is within an additive $\beta$ of $u$ so long as one correct collision holds.

Continuing, we next lower bound the probability of one correct collision. For a fixed $g_{i_w l} \in G_{i_w}$, we have that $g(w) = g(u)$ iff $h_{i_w l k}(w) = h_{i_w l k}(u)$ for all $k \leq K$. The probability of this for one $k \leq K$ is, by definition, $p_1$. By Theorem 6 we know that $\frac{\log(1/p_1)}{\log(1/p_2)} = 1/c^2 + o(1)$ and so for a fixed $l$, we have that this occurs with probability at least

$$\begin{aligned}
p_1^K &= \exp(-K \cdot \log(1/p_1)) \\
&= \exp\left(-\frac{\log(1/p_1)}{\log(1/p_2)} \cdot (O(\log n) + \log L + \log\log(\Delta/\beta))\right) \\
&= \exp(-(1/c^2 + o(1)) \cdot (O(\log n) + \log L + \log\log(\Delta/\beta))) \\
&\geq \exp\left(-(1/c^2 + o(1)) \cdot O(\log n)\right)
\end{aligned}$$

It follows that the probability that this does not hold for some $l$ is at most

$$\begin{aligned}
\left(1 - p_1^K\right)^L &\leq \exp(-L \cdot p_1^K) \\
&\leq \exp\left(-L \cdot \exp\left[-(1/c^2 + o(1)) \cdot O(\log n)\right]\right) \\
&= \exp\left(-\Omega(\log n) - \exp[((1/c^2) + o(1)) \cdot \Omega(\log n)] \cdot \exp\left[-(1/c^2 + o(1)) \cdot O(\log n)\right]\right) \\
&= n^{-\Omega(1)}.
\end{aligned}$$

Next, we lower bound the probability of no incorrect collisions. Fix an $i < i_w$. Observe that for a given $h_{ilk}$ we have, by definition, that $h_{ilk}(u) = h_{ilk}(w)$ with probability at most $p_2$. Thus, we have $g_{il}(u) = g_{il}(w)$ with probability at most

$$p_2^K = \exp(-\log(1/p_2)K) = \exp(-\Omega(\log n) - \log L - \log\log(\Delta/\beta)).$$

Union bounding over our at most $\log(\Delta/\beta)$-many such $i$ and $L$-many such $l$, we get that that we have no incorrect collisions except with probability at most

$$p_2^K \cdot \log(\Delta/\beta) \cdot L = \exp(-\Omega(\log n)) = n^{-\Omega(1)}.$$

Finally, union bounding over our $n$-many possible points in $S$ and using the fact that $\gamma \geq 1$ we have that we succeed except with probability at least a fixed constant bounded away from $0$ for $S$ and this query. Furthermore, by taking $\gamma$-many independent repetitions of this data structure and taking the returned point closest to our query, we increase our runtime by a multiplicative $\gamma$ and reduce our failure probability to $\exp(-\gamma)$. $\qquad\square$

## C  Proofs from Section 3

**Lemma 1.** *Given $s \in \mathbb{R}^d$ and $\beta, \Delta > 0$, there exists a* covering net *$Q \subseteq \mathbb{R}^d$ such that*

1. ***Small Size:*** *$|Q| \leq (\Delta/\beta)^{O(d\log d)}$*
2. ***Queries:*** *given $u \in B_s(\Delta)$, one can compute $u' \in Q$ in time $O(d)$ such that $D(u, u') \leq \beta$.*

*Proof.* The basic idea is simply to take an evenly spaced hypergrid centered at $s$ of radius $\Delta$. More formally, we let

$$Q := \left\{ s + \frac{\beta}{\sqrt{d}} \cdot y : y \in \mathbb{Z}^d \cap \left( \left[ -\Delta\frac{\sqrt{d}}{\beta}, \Delta\frac{\sqrt{d}}{\beta} \right] \right)^d \right\}$$

where above $\left( \left[ -\Delta\frac{\sqrt{d}}{\epsilon}, \Delta\frac{\sqrt{d}}{\beta} \right] \right)^d$ is the $d$-way Cartesian product. That is, $y$ is a $d$-dimensional vector whose coordinates are integers between $-\Delta\frac{\sqrt{d}}{\beta}$ and $\Delta\frac{\sqrt{d}}{\beta}$. Thus, $Q$ consists of all points which offset each coordinate of $s$ by a multiple of $\frac{\beta}{\sqrt{d}}$ up to total offset distance $\Delta$ in each coordinate.

The number of points in $Q$ is trivially the number of offsets $y$ which is, in turn,

$$\left( 2\frac{\Delta \cdot \sqrt{d}}{\beta} \right)^d = \left( \frac{\Delta}{\beta} \right)^{O(d\log d)}$$

as desired.

Next we analyze how to query. Given a point $u$, we let

$$u_i' := s_i + \frac{\beta}{\sqrt{d}} \cdot \left\lfloor \frac{\sqrt{d}}{\beta}(u_i - s_i) \right\rfloor$$

be $u_i$ rounded down to the nearest multiple of $\frac{\beta}{\sqrt{d}}$ after offsetting by $s_i$. Observe that $u' = (u_1', u_2', \dots u_d') \in Q$ by our assumption that $u \in B_s(\Delta)$ and that, furthermore, $u'$ can be computed from $u$ in time $O(d)$. Lastly, observe that

$$|u_i - u_i'| \leq \frac{\beta}{\sqrt{d}}$$

and so we have that the distance between $u$ and $u'$ is

$$D(u, u') = \sqrt{\sum_i (u_i - u_i')^2} \leq \sqrt{\sum_i \left( \frac{\beta}{\sqrt{d}} \right)^2} = \beta.$$

$\qquad\square$

**Lemma 2.** *Algorithm 1 is a $(c, c\beta)$-approximate nearest-neighbor search data structure with high probability for adaptive updates.*

*Proof.* Consider a query on $(u, \bar{u})$ with corresponding $u' \in Q$ in our covering net (i.e., the $u'$ returned by Lemma 1 when input $u$). We refer to the state of our data structure at the beginning of its for loop (i.e., upon receiving a new update or query) as at the beginning of an iteration.

First, observe that a simple argument by induction on our insertions and deletions shows that the point set for which $\mathcal{N}_i$ is a data structure at the beginning of an iteration is $S_i$ and that, furthermore, if $S$ is the set of points which have been inserted but not deleted at a given point in time then $\{S_i\}_i$ partitions $S$. Thus, we have

$$\min_i D(u', S_i \setminus \bar{u}) = D(u', S \setminus \bar{u}).$$

Furthermore, by the guarantees of Lemma 1 and the fact that all points of $S$ and all query points are always contained in $B_s(\Delta)$, we know that

$$D(u, u') \leq O(\beta). \tag{3}$$

It follows that

$$D(u', S \setminus \bar{u}) \leq D(u, S \setminus \bar{u}) + O(\beta)$$

and so

$$\min_i D(u', S_i \setminus \bar{u}) \leq D(u, S \setminus \bar{u}) + O(\beta). \tag{4}$$

Furthermore, notice that since the randomness of $\mathcal{N}_i$ is chosen after we fix $S_i$, we have by assumption that for a fixed $u' \in Q$ that $\mathcal{N}_i$ succeeds for a query on $(u', \bar{u})$ with points $S_i$ except with probability at most

$$n^{-\Omega(d \log d \cdot \log(\Delta/\beta))}.$$

On the other hand, by Lemma 1 we know that $|Q| \leq \left(\frac{\Delta}{\beta}\right)^{O(d \log d)}$ so taking a union bound over all points in $Q$ we have that with high probability $\mathcal{N}_i$ succeeds for $S_i$ and every query point in $Q$ except with probability at most $1/\mathsf{poly}(n)$. Union bounding over our $O(n)$-many $\mathcal{N}_i$ that we ever instantiate, we get that every $\mathcal{N}_i$ succeeds for its corresponding $S_i$ and every query point of $Q$ except with probability at most $1/\mathsf{poly}(n)$. In other words, with high probability we always have

$$D(u', v_i) \leq c \cdot D(u', S_i \setminus \bar{u}) + O(\beta). \tag{5}$$

Thus, combining the triangle inequality, Equations 3, 4 and 5, we have that with high probability whenever we query point $u$, the returned point $v$ satisfies

$$\begin{aligned} D(u, v) &= \min_i D(u, v_i) \\ &\leq D(u, u') + \min_i D(u', v_i) \\ &\leq O(\beta) + \min_i D(u', v_i) \\ &\leq O(\beta) + c \cdot \min_i \cdot D(u', S_i \setminus \bar{u}) \\ &\leq c \cdot D(u, S \setminus \bar{u}) + O(c \cdot \beta) \end{aligned}$$

Choosing our hidden constant appropriately, we get that with high probability our data structure is indeed $(c, c\beta)$-approximate. $\qquad\square$

**Lemma 3.** *Algorithm 1 has amortized insertion time $O(\log n) \cdot T_I$ where $T_I$ is the insertion time of the input oblivious dynamic ANNS data structure.*

*Proof.* Let $n_{\text{INSERT}} \leq n$ be the total number of insertions up to some point in time. Each time we instantiate an $\mathcal{N}_{i+1}$ on an $S_{i+1}$, it consists of at most $2^{i+2}$-many points and so by assumption takes us time

$$T_I \cdot 2^{(i+2)}$$

time to instantiate. On the other hand, the total number of times we can instantiate $\mathcal{N}_{i+1}$ is at most $n_{\text{INSERT}}/2^i$ (since each time we instantiate it we add a new set of at least $2^i$ points to $S_{i+1}$ and points only move from $S_j$s to $S_{j+1}$s). It follows that the total time to instantiate $\mathcal{N}_{i+1}$ is at most

$$T_I \cdot 2^{(i+2)} \cdot \frac{n_{\text{INSERT}}}{2^i} = O(T_I \cdot n_{\text{INSERT}}).$$

Applying, the fact that $i \leq \log n$, we get that the total time for all insertions is

$$O(\log n) \cdot T_I \cdot n_{\text{INSERT}}.$$

Dividing by our $n_{\text{INSERT}}$-many insertions, we get an amortized insertion time of $O(\log n) \cdot T_I$, as desired. $\qquad\square$

**Theorem 2** (Reduction of Dynamic ANNS from Oblivious to Adaptive). *Suppose we are given $c$, $\beta$, $\Delta$, $n$, $s \in \mathbb{R}^d$ and dynamically updated set $S \subset \mathbb{R}^d$ with at most $n$ insertions, such that all inserted points and query points lie in $B_s(\Delta)$. Moreover, assume that we can compute a dynamic $(c, \beta)$-approximate NNS data structure with query, deletion and insertion times $T_Q$, $T_D$ and $T_I$ which succeeds for $S$ and a fixed query except with probability at most $n^{-O(d \log d \log(\Delta/\beta))}$.*

*Then, we can compute a randomized dynamic $(c, c\beta)$-approximate NNS data structure that succeeds for adaptive updates with query, deletion and amortized insertion times $O(\log n) \cdot T_Q$, $O(T_D)$ and $O(\log n) \cdot T_I$.*

*Proof.* We use Algorithm 1. Correctness follows from Lemma 2 and the amortized insertion time follows from Lemma 3. The deletion time is immediate from the fact that we can compute the $i$ such that $u \in S_i$ in constant time. Lastly, we analyze our query time. By Lemma 1, computing each $u'$ takes time $O(d) \leq T_Q$ (since reading the query takes time $\Omega(d)$). Likewise, by assumption, computing $\mathcal{N}_i^{\text{QUERY}}(u', \bar{u})$ takes time $T_Q$ for each $i$, giving our desired bounds. $\qquad\square$

# D   Proofs from Section 4

We now prove that Algorithm 3 gives a $(c(1 + \epsilon))$-approximate HAC where $c$ is the approximation value inherited from the approximate nearest neighbor data structure. Note that a smaller $\epsilon$ will give a more accurate HAC but will increase the runtime of the algorithm.

**Lemma 4.** *Algorithm 3 gives a $c(1 + \epsilon)$-approximate Centroid-Linkage HAC.*

*Proof.* The idea of the proof is to observe that one of the two endpoints of the closest pair will always store an approximate of the minimum distance in $Q$. In that spirit, suppose that right before a centroid is dequeued, the shortest distance between active centroids is $l_{\text{OPT}}$. Then there must exist active centroids $x_1$ and $x_2$ such that $D(x_1, x_2) = l_{\text{OPT}}$. Assume without loss of generality that $x_1$ was queued before $x_2$ and $x_2$ was queued with distance $l_2$. Then, when $x_2$ was queued, $x_1$ was active so $l_2 \leq c \cdot l_{\text{OPT}}$.

Thus, when a centroid, $x$, is dequeued, its distance $l_x$ obeys the inequality

$$l_x \leq l_2 \leq c \cdot l_{\text{OPT}}.$$

Then, if $x$ is merged during the while loop, it is merged with a centroid of distance at most

$$(1 + \epsilon)l_x \leq c(1 + \epsilon)l_{\text{OPT}}$$

as desired. If a centriod gets merged without being dequeued, that means it merged with another identical centroid which is consistent with 1-approximate HAC. $\qquad\square$

We now prove the runtime of the algorithm. Say that the maximum pairwise distance between points in $P$ is $\Delta$ and the minimum is $\delta$. We make the following observations about the geometry of the centriods throughout the runtime of Algorithm 3.

**Observation 1.** *Consider any point $p \in P$. Throughout the runtime of Algorithm 3, the distance from any centroid to $p$ is at most $\Delta$.*

*Proof.* Consider a cluster $Q \subseteq P$. Since for each $q \in Q$ we have $D(p, q) \leq \Delta$, it follows that $D(p, \text{cent}(Q)) \leq \Delta$. $\qquad\square$

**Observation 2.** *When running Algorithm 3, at any point in time there can only ever be a single centroid queued with distance less than $\delta$.*

*Proof.* Initially, every centroid is queued with distance at least $\delta$. Thus, if a centroid is ever queued with a smaller distance it will immediately be dequeued and merged. $\square$

**Lemma 5.** *Algorithm 3 runs in time $O(n \cdot T_I + n \cdot T_D + n \cdot T_Q \cdot \log_{1+\epsilon}(\frac{2\Delta}{\delta}) + n \log(n) \log_{1+\epsilon}(\frac{2\Delta}{\delta}))$.*

*Proof.* The idea of the proof is that each time we dequeue a centroid we either merge it or we increase the distance associated with it by a multiplicative factor of $1 + \epsilon$ which can only happen $\log_{1+\epsilon}(2\Delta/\delta)$ times for each of our $O(n)$-many centroids.

Each centroid is inserted into $\mathcal{N}$ at most once, either during the initial construction or in Algorithm 2, and deleted at most once. Since there are $2n - 1$ centroids throughout the runtime, the total time spent on updates to $\mathcal{N}$ is $O(n \cdot T_I + n \cdot T_D)$.

At any given point during the algorithm, a centroid can be in $Q$ at most once. Since only $2n - 1$ centroids are ever created, there are at most $2n - 1$ elements in $Q$ at any time. Thus, a single queue or dequeue operation takes $O(\log(n))$ time. In the initial for loop we queue $n$ elements to $Q$ and make $n$ queries to $\mathcal{N}$ taking time $O(n \log(n) + n \cdot T_Q)$. Each time algorithm 2 gets called it queues one element to $Q$ and makes up to 4 updates to $C$. Since Algorithm 2 is called at most $n - 1$ times, the total time spent on merges, not including updates to $\mathcal{N}$, is $O(n \cdot T_Q + n \log(n))$.

Lastly, we consider how many times a centroid $x$ can be dequeued on Line 8. Suppose $x$ is queued with distance $l_i$ the $i$th time it is queued. If $l_1 < \delta$ then by Observation 2, $x$ will be immediately dequeued and merged so $x$ is only queued once. Thus, we assume $l_1 \geq \delta$. If $x$ is dequeued with distance $l_i$ then either $x$ is merged and will not be queued again or it is queued again with distance

$$l_{i+1} > (1 + \epsilon)l_i > (1 + \epsilon)^i l_1.$$

Since $l_1 \geq \delta$ by assumption and $l_i \leq 2\Delta$ for all $i \geq 1$ by Observation 1, it follows that $x$ can only be queued, and therefore dequeued, $\log_{1+\epsilon}(\frac{2\Delta}{\delta})$ times. Each time, not including merges, there can be one dequeue from $Q$, one queue to $Q$, and one query to $\mathcal{N}$. Since there are $2n - 1$ centroids created by the algorithm, the while loop (not including merges) runs in time $O(n \log(n) \log_{1+\epsilon}(\frac{2\Delta}{\delta}) + n \cdot T_Q \cdot \log_{1+\epsilon}(\frac{2\Delta}{\delta}))$. $\square$

Combining the previous two lemmas together we get the following.

**Theorem 4** (Reduction of Centroid HAC to Dynamic ANN for Adaptive Updates). *Suppose we are given a set of $n$ points $P \subset \mathbb{R}^d$, $c > 1$, $\epsilon > 0$, and lower and upper bounds on the minimum and maximum pairwise distance in $P$ of $\delta$ and $\Delta$ respectively. Suppose we are also given a data structure that is a dynamic $c$-approximate NNS data structure for all queries in Algorithm 3 that works for adaptive updates with insertion, deletion and query times of $T_I$, $T_D$, and $T_Q$.*

*Then there exists an algorithm for $c(1 + \epsilon)$-approximate Centroid-Linkage HAC on $n$ points that runs in time $\tilde{O}(n \cdot (T_I + T_D + T_Q))$ assuming $\frac{\Delta}{\delta} \leq \mathsf{poly}(n)$ and $\frac{1}{\epsilon} \leq \mathsf{polylog}(n)$.*

*Proof.* By Lemma 4 and Lemma 5 we get that Algorithm 3 gives $c(1 + \epsilon)$-approximate HAC in time $O(n \cdot T_I + n \cdot T_D + n \cdot T_Q \cdot \log_{1+\epsilon}(\frac{2\Delta}{\delta}) + n \log(n) \log_{1+\epsilon}(\frac{2\Delta}{\delta}))$. It is only left to show that $\log_{1+\epsilon}(\frac{2\Delta}{\delta}) \leq \mathsf{polylog}(n)$. Changing the base of the logarithm we get

$$\log_{1+\epsilon}\left(\frac{2\Delta}{\delta}\right) = \frac{\log(\frac{2\Delta}{\delta})}{\log(1 + \epsilon)}.$$

We have that $\log(\frac{2\Delta}{\delta}) \leq \mathsf{polylog}(n)$ by the assumption that $\frac{\Delta}{\delta} \leq \mathsf{poly}(n)$. By the inequality $\log(1 + x) \geq \frac{x}{1+x}$ for all $x > -1$ we get that if $\epsilon \leq 1$

$$\frac{1}{\log(1 + \epsilon)} \leq \frac{1 + \epsilon}{\epsilon} \leq \frac{2}{\epsilon} = O\left(\frac{1}{\epsilon}\right) \leq \mathsf{polylog}(n)$$

where the final inequality comes from the assumption that $\frac{1}{\epsilon} \leq \mathsf{polylog}(n)$. If $\epsilon > 1$ then

$$\frac{1}{\log(1+\epsilon)} \leq \frac{1}{\log(2)} \leq \mathsf{polylog}(n).$$

Combining the above inequalities and the above-stated runtime gives our final runtime. $\qquad \square$

We use Theorem 4 with our results from Section 3 to get our main result. First we prove the following lemma which says that our ANNS that includes an additive term functions as a purely multiplicative ANNS for our algorithm.

**Lemma 6.** *For any $\lambda \in (1, c)$, if $\mathcal{N}$ is a $\left( \frac{c}{\lambda}, \frac{\delta(\lambda-1)}{(1+c)\lambda} \right)$-approximate NNS data structure that works for adaptive updates then $\mathcal{N}$ is a c-approximate NNS data structure for all queries in Algorithm 3.*

*Proof.* Let $C$ be the set of active centroids at the time any given query takes place. We will say a point $x$ is close to the set $C$ if $D(x, C) < \frac{\delta}{c}$. We show that if $x$ is not close to $C$, then $\mathcal{N}^{\mathrm{query}}(x, x) = y$ will give us the multiplicative result we desire:

$$
\begin{aligned}
D(x, y) &\leq \frac{c}{\lambda} D(x, C) + \frac{\delta(\lambda - 1)}{(1 + c)\lambda} \\
&\leq \frac{c}{\lambda} D(x, C) + \frac{\delta}{c}(\lambda - 1) \\
&\leq \frac{c}{\lambda} D(x, C) + D(x, C)(\lambda - 1) \\
&< \frac{c}{\lambda} D(x, C) + D(x, C)\frac{c(\lambda - 1)}{\lambda} \\
&\leq c \cdot D(x, C)\left( \frac{1}{\lambda} + \frac{\lambda - 1}{\lambda} \right) \\
&\leq c \cdot D(x, C). \qquad\qquad\qquad\qquad\qquad (6)
\end{aligned}
$$

We now prove the lemma by induction. The base case is the set of neighbors calculated for the initial set of points $P$. Each $p \in P$ is not close to $C \setminus p$ so by Eq. (6),

$$D(p, \mathcal{N}^{\mathrm{query}}(p, p)) < c \cdot D(p, C \setminus p)$$

as desired. Now for the induction step, consider some query for a centroid $x$ that has just been dequeued. For $x, y \in C$, say $y$ is a $c$-approximate nearest neighbor of $x$ if $D(x, y) \leq c \cdot D(x, C \setminus x)$. Assume by induction that all queries up to this point have returned $c$-approximate nearest neighbors. We know by Observation 2 and the inductive hypothesis that either no points are close to $C$ or $x$ is the lone point that is close to $C$. In the former case, $\mathcal{N}$ returns a $c$-approximate nearest neighbor by Eq. (6). Now consider the latter case. Let $y$ be a centroid such that $D(x, y) > c \cdot D(x, C \setminus x)$. We will show that $\mathcal{N}^{\mathrm{query}}(x, x) \neq y$.

Define $L = D(x, y) + D(x, C \setminus x)$. By the triangle inequality and because $x$ is the only point that is close to $C$, we have $L \geq \frac{\delta}{c}$. Define $p_y = D(x, y)/L$ and $p_S = D(x, C \setminus x)/L$ to be the percentage of the distance $L$ from $D(x, y)$ and $D(x, C \setminus x)$ respectively. It is enough for us to show that

$$\frac{c}{\lambda}(p_S \cdot L) + \frac{\delta(\lambda - 1)}{(1 + c)\lambda} < p_y \cdot L$$

for all $L \geq \frac{\delta}{c}$. Since this equation is linear in $L$, if it is true for any such $L$ then it is true for $L = \frac{\delta}{c}$. Since $p_y > c \cdot p_S$, we have $p_S < \frac{1}{1+c}$ and $p_y > \frac{c}{1+c}$. Plugging in $L = \frac{\delta}{c}$ we get

$$\frac{c}{\lambda}\Big(p_S \cdot L\Big) + \frac{\delta(\lambda - 1)}{(1 + c)\lambda} < \frac{c}{\lambda}\Big(\frac{1}{1 + c} \cdot \frac{\delta}{c}\Big) + \frac{\delta(\lambda - 1)}{(1 + c)\lambda}$$

$$= \frac{\delta + \delta(\lambda - 1)}{(1 + c)\lambda}$$

$$= \frac{\delta}{(1 + c)}$$

$$< p_y \cdot \frac{\delta}{c}$$

$$= p_y \cdot L$$

as desired. $\qquad\square$

**Theorem 5** (Centroid HAC Algorithm)**.** *For $n$ points in $\mathbb{R}^d$, there exists a randomized algorithm that given any $c > 1.01$ and lower and upper bounds on the minimum and maximum pairwise distance in $P$ of $\delta$ and $\Delta$ respectively, computes a $c$-approximate Centroid-Linkage HAC with high probability. It has runtime $O(n^{1 + O(1/c^2)} d^2)$ assuming $c, d, \frac{\Delta}{\delta} \leq \mathsf{poly}(n)$.*

*Proof.* Let $\epsilon = 0.001$ and define $\hat{c} = \frac{c}{1+\epsilon} > 1.005$. For any $\lambda \in (1, \hat{c})$, Theorem 3 says we can construct a $\Big(\frac{\hat{c}}{\lambda}, \frac{\delta(\lambda - 1)}{(1 + \hat{c})\lambda}\Big)$-approximate nearest neighbor data structure, $\mathcal{N}$, with query, deletion, and amortized insertion time $n^{O(\lambda^2/\hat{c}^2)} \log\Big(\frac{\Delta(1 + \hat{c})\lambda}{\delta(\lambda - 1)}\Big) d^2$. Note that we moved the big $O$ in Theorem 3 from the approximation guarantee to the runtime by scaling by some constant. By Lemma 6, $\mathcal{N}$ is a $\hat{c}$-approximate ANNS data structure for all queries in Algorithm 3. By Theorem 4 and our definition of $\hat{c}$, we get $c$-approximate centroid HAC by running Algorithm 3 with $\mathcal{N}$. The runtime is $\tilde{O}(n^{1 + O(\lambda^2/\hat{c}^2)} \log\Big(\frac{\Delta(1 + \hat{c})\lambda}{\delta(\lambda - 1)}\Big) d^2)$. Letting $\lambda = 1.005$ and by the assumptions that $\frac{\Delta}{\delta}, c \leq \mathsf{poly}(n)$, all $\log$ terms are dominated by the big $O$ in the exponent and we get the runtime $O(n^{1 + O(1/\hat{c}^2)} d^2) = O(n^{1 + O(1/c^2)} d^2)$. $\qquad\square$

In Algorithm 3, in the worst case we may have to queue a centroid $\log_{1+\epsilon}(\frac{2\Delta}{\delta})$ times. However, in practice a centroid may only be queued a small number of times. Say a tuple $(l, x, y)$ in our priority queue becomes stale if the centroid $y$ is merged. Let $\Gamma$ be the average number of tuples in our priority queue that become stale with each merge. Our experiments summarized in Table 4 suggest that $\Gamma$ is a small constant in practice with $\Gamma < 1$ for all experiments. The following lemma shows that in this case we lose the $\log_{1+\epsilon}(\frac{2\Delta}{\delta})$ in the runtime.

**Lemma 7.** *If the average number of queued element made stale for each merge in Algorithm 3 is $\Gamma$, then Algorithm 3 runs in time $O(n \cdot T_I + n \cdot T_D + n \cdot T_Q \cdot \Gamma + n \log(n) \cdot \Gamma)$.*

*Proof.* There are a total of $n - 1$ merges during the runtime of Algorithm 3 so the total number of tuples that are made stale is $(n - 1)\Gamma$. There are an additional $n - 1$ tuples that get merged. Thus, the total number of tuples that get dequeued throughout the runtime of the algorithm is $O(\Gamma)$. The rest of the proof is the same as that of Lemma 5 except we pay the cost of de-queuing on Line 8 $O(n \cdot \Gamma)$ times instead of $n \cdot \log_{1+\epsilon}(\frac{2\Delta}{\delta})$. $\qquad\square$

## E   Empirical Evaluation

**Experimental Setup:** We run our experiments on a 96-core Dell PowerEdge R940 (with two-way hyperthreading) machine with $4 \times 2.4\text{GHz}$ Intel 24-core 8160 Xeon processors (with 33MB L3 cache) and 1.5TB of main memory. Our programs are implemented in C++ and compiled with the g++ compiler (version 11.4) with $\texttt{-O3}$ flag.

| Dataset | n | d | # Clusters |
|---|---|---|---|
| iris | 150 | 4 | 3 |
| wine | 178 | 13 | 3 |
| cancer | 569 | 30 | 2 |
| digits | 1797 | 64 | 10 |
| faces | 400 | 4096 | 40 |
| mnist | 70000 | 784 | 10 |
| birds | 84635 | 1024 | 525 |
| covertype | 581012 | 54 | 7 |
| sift-1M | 1000000 | 128 | - |

Table 2: Datasets

**Datasets** The details of the various datasets used in our experiments are stated in Table 2. mnist[13] is a standard machine learning dataset that consists of $28 \times 28$ dimensional grayscale images of digits between 0 and 9. Here, each digit corresponds to a cluster. The birds dataset contains $224 \times 224 \times 3$ dimensional images of 525 species of birds. As done in previous works [42], we pass each image through ConvNet [27] to obtain an embedding. The ground truth clusters correspond to each of the 525 species of birds. The licenses of these datasets are as follows: iris (CC BY 4.0), wine (CC BY 4.0), cancer (CC BY 4.0), digits (CC BY 4.0), faces (CC BY 4.0), Covertype (CC BY 4.0), mnist (CC BY-SA 3.0 DEED), birds (CC0: Public Domain), and sift-1M (CC0: Public Domain).

**Algorithms Implemented** We implement the approximate Centroid HAC algorithm described in Algorithm 3, using the dynamic ANNS implementation based on DiskANN [38, 18]. We denote this algorithm as $(1 + \epsilon)$-Centroid HAC or Centroid$_\epsilon$. We emphasize that although we refer to this as $(1 + \epsilon)$-Centroid HAC, it is not necessarily a *true* $(1 + \epsilon)$-approximate algorithm since we are using a heuristic ANNS data structure with no theoretical guarantees on the approximation factor. The use of $(1 + \epsilon)$ here captures only the loss from merging a "near-optimal" pair of clusters, as detailed in Algorithm 3.

As our main baseline, we consider the optimized implementation of exact Centroid HAC from the fastcluster[4] package. This implementation requires quadratic space since it maintains the distance matrix, and is a completely sequential algorithm. We also implement an efficient version of exact Centroid HAC based on our framework (i.e. setting $\epsilon = 0$ and using exact NNS queries) which has the benefit of using only linear space and supports parallelism in NNS queries.

We also implement a *bucket-based* version of approximate Centroid HAC as a baseline, based on the approach of [1]. The algorithm of [1] runs in rounds processing pairs of clusters whose distance is within the threshold defined by that round; the threshold value scales by a constant factor, resulting in logarithmic (in aspect ratio) number of rounds. In each round, they consider a pair of clusters that are within the threshold distance away, and merge them. For the linkage criteria considered in [1], the distances between two clusters can only go up as a result of two clusters merging. However, this is not true for Centroid HAC, as discussed in Section 1.

Here, we observe that when two clusters merge in a round, the only distances affected are distances of other clusters to the new merged cluster. Thus, we can repeatedly check if the nearest neighbor of this cluster is still within the threshold and merge with it in that case. Thus, in a nutshell, the algorithm filters clusters whose nearest neighbors are within the current threshold in each round and merges clusters by the approach mentioned above. This algorithm requires many redundant NNS queries, as we will see during the experimental evaluation.

### E.1 Quality Evaluation

The output of a hierarchical clustering algorithm is typically a tree of clusters, called a *dendrogram*, that summarizes all the merges performed by the algorithm. The leaves correspond to the points in the dataset, and each internal node represents the cluster formed by taking all the leaves in its subtree. There is a cost associated to each internal node denoting the cost incurred when merging two clusters to form the cluster associated to that node. Given a threshold value, a clustering is obtained by cutting the dendrogram at certain internal nodes whose associated cost is greater than the threshold.

---

[4]https://pypi.org/project/fastcluster/

Table 3: The ARI, NMI, Dendrogram Purity and Dasgupta Cost of our approximate Centroid HAC implementations for $\epsilon = 0.1, 0.2, 0.4$, and $0.8$, versus Exact Centroid HAC. The best quality score for each dataset is in bold and underlined.

| | Dataset | Centroid$_{0.1}$ | Centroid$_{0.2}$ | Centroid$_{0.4}$ | Centroid$_{0.8}$ | Exact Centroid |
|---|---|---|---|---|---|---|
| **ARI** | iris | **0.759** | 0.746 | 0.638 | 0.594 | **0.759** |
| | wine | 0.352 | 0.352 | **0.402** | 0.366 | 0.352 |
| | cancer | 0.509 | 0.526 | 0.490 | **0.641** | 0.509 |
| | digits | 0.589 | 0.571 | 0.576 | **0.627** | 0.559 |
| | faces | 0.370 | 0.388 | **0.395** | 0.392 | 0.359 |
| | mnist | **0.270** | 0.222 | 0.218 | 0.191 | 0.192 |
| | birds | 0.449 | 0.449 | 0.442 | **0.456** | 0.441 |
| | **Avg** | **0.471** | 0.465 | 0.452 | 0.467 | 0.453 |
| **NMI** | iris | **0.803** | 0.795 | 0.732 | 0.732 | **0.803** |
| | wine | **0.424** | **0.424** | 0.413 | 0.389 | **0.424** |
| | cancer | 0.425 | 0.471 | 0.459 | **0.528** | 0.425 |
| | digits | 0.718 | 0.726 | 0.707 | **0.754** | 0.727 |
| | faces | 0.539 | 0.534 | 0.549 | 0.549 | **0.556** |
| | mnist | 0.291 | 0.282 | 0.306 | **0.307** | 0.250 |
| | birds | 0.748 | 0.747 | 0.756 | **0.764** | 0.743 |
| | **Avg** | 0.564 | 0.569 | 0.560 | **0.575** | 0.561 |
| **Purity** | iris | 0.869 | 0.862 | 0.808 | 0.809 | **0.871** |
| | wine | 0.616 | 0.616 | **0.640** | 0.601 | 0.616 |
| | cancer | 0.816 | 0.818 | 0.805 | **0.836** | 0.816 |
| | digits | 0.677 | 0.667 | 0.654 | **0.715** | 0.679 |
| | faces | 0.460 | 0.477 | 0.478 | **0.488** | 0.467 |
| | mnist | **0.310** | 0.286 | 0.283 | 0.278 | 0.308 |
| | birds | 0.555 | 0.550 | 0.543 | 0.516 | **0.559** |
| | **Avg** | 0.615 | 0.611 | 0.602 | 0.606 | **0.616** |
| **Dasgupta** | iris | 506011.1 | 506101.2 | 510680.6 | 507149.4 | **505809.8** |
| | wine | 7655.9 | 7655.9 | **7545.7** | 7564.2 | 7655.9 |
| | cancer | 153644.7 | **153519.2** | 156220.4 | 156893.8 | 153843.2 |
| | digits | 39292919.2 | 39315744.3 | 39299184.6 | **39215081.5** | 39289534.3 |
| | faces | 1703728.2 | 1703696.7 | 1706886.4 | **1703361.3** | 1704833.1 |
| | mnist | $11193\times10^9$ | $11195\times10^9$ | $11194\times10^9$ | $\mathbf{11185\times10^9}$ | $11195\times10^9$ |
| | birds | $5198\times10^9$ | $5199\times10^9$ | $5196\times10^9$ | $\mathbf{5183\times10^9}$ | $5201\times10^9$ |
| | **Avg** | $2341\times10^9$ | $2342\times10^9$ | $2341\times10^9$ | $\mathbf{2338\times10^9}$ | $2342\times10^9$ |

In our experiments, the ARI and NMI scores are calculated by taking the best score obtained over the clusterings formed by considering all possible *thresholded*-cuts to the output dendrogram. For larger datasets, we consider cuts only at threshold values scaled at a constant factor (i.e. logarithmic-many cuts).

Table 3 contains the results obtained for our quality evaluations. We also plot the scores obtained by approximate Centroid HAC as a function of $\epsilon$; see Figure 6, Figure 7, Figure 8, Figure 9, and Figure 10.

We observe that the quality of the clustering produced by approximate Centroid HAC is generally comparable to that of the exact Centroid HAC algorithm. On average, the quality is slightly better in some cases, but we attribute this to noise. In particular, we observe that for the value of $\epsilon = 0.1$, we consistently get comparable quality to that of exact centroid: the ARI and NMI scores are on average within a factor of 7% and 2% to that of exact Centroid HAC, respectively. We also obtained good dendrogram purity score and Dasgupta cost as well, with values within 0.3% and 0.03%, respectively.

**Non-Monotonicity of CentroidHAC** Unlike other HAC algorithms, the costs of merging clusters for centroid HAC is not monotone (i.e., non-decreasing). The monotonicity property of HAC algorithms is useful when producing the clusterings for applications from the output dendrgoram: typically the dendrogram is cut at certain thresholds, and the disconnected subtrees obtained are flattened to compute the clusters. Here, cutting the dendrogram for a given threshold returns a well-defined and

unique clustering. However, this is not the case with Centroid HAC. Indeed, for a given threshold value, we can obtain different clusterings depending on how we define to cut the dendrogram. For e.g., if two nodes $U$ and $V$ exists in the output dendrogram with associated merge costs (i.e., cost incurred at the time of merging the two clusters to form that node) of $U$ being greater than $V$, and $V$ is an ancestor of $U$ (which is a possibility for Centroid HAC). Then, given a threshold value $cost(V) < \tau < cost(V)$, we could either cut at $V$ (or above), or stop somewhere below $U$.

We would like to characterize and study this non-monotonicity of Centroid HAC in real-world datasets. For this, we define the following notion: for nodes $U, V$ in the output dendrogram, the pair $(U, V)$ is called a $\delta$-*inversion* if $V$ is an ancestor of $U$ and $cost(U) \geq (1 + \delta)cost(V)$, where $cost(U)$ is the merge cost associated to node $U$.

We calculate the number of such inversions incurred by exact Centroid HAC, and compare with the inversions incurred by our approximate algorithm. Figure 5 shows the number of $\delta$-inversions incurred on the benchmark datasets. We observe that for small values of $\epsilon$, approximate Centroid HAC has similar number of $\delta$ inversions as that of exact Centroid HAC.

### E.2 Running Time Evaluation

Figures 4a and 4b shows the running times on varying slices of the SIFT-1M dataset using one thread and 192 parallel threads, respectively. For the approximate methods, we considered $\epsilon = 0.1$. Figure 4c further compares the heap and bucket based approaches as a function of $\epsilon$.

We observe that the bucket based approach is very slow for small values of $\epsilon$ due to many redundant nearest-neighbor computations. Yet, at $\epsilon = 0.1$, both algorithms, with 192 threads, obtain speed-ups of up to $175\times$ with an average speed-up of $22\times$ compared to the exact counterparts. However, on a single thread, the bucket based approach fails to scale, while the heap based approach still achieves significant speed-ups of upto $36\times$ with an average speed-up of $5\times$. Overall, our heap-based approximate Centroid HAC implementation with $\epsilon = 0.1$ demonstrates good quality and scalability, making it a good choice in practice.

**Stale Merges and Number of Queries** We compute the number of *stale merges* (see Section 1) incurred by our algorithm on various benchmark datasets. See Table 4. We observe that, on average, the number of such stale queries are incurred about 60% of the time compared to the number of actual merges (i.e. number of points).

<table>
<tr><td colspan="3">Table 4: Number of Stale Merges</td><td colspan="3">Table 5: Number of ANNS Queries</td></tr>
<tr><td>Dataset</td><td>$n$</td><td>Centroid$_{0.1}$</td><td>Dataset</td><td>Centroid$_{0.1}$(Heap)</td><td>Centroid$_{0.1}$(Bucket)</td></tr>
<tr><td>iris</td><td>150</td><td>123</td><td>iris</td><td>561</td><td>4112</td></tr>
<tr><td>wine</td><td>178</td><td>161</td><td>wine</td><td>686</td><td>7011</td></tr>
<tr><td>cancer</td><td>569</td><td>497</td><td>cancer</td><td>2167</td><td>24852</td></tr>
<tr><td>digits</td><td>1797</td><td>875</td><td>digits</td><td>5583</td><td>48523</td></tr>
<tr><td>faces</td><td>400</td><td>90</td><td>faces</td><td>1000</td><td>9862</td></tr>
<tr><td>mnist</td><td>70000</td><td>33039</td><td>mnist</td><td>221086</td><td>2481831</td></tr>
<tr><td>birds</td><td>84635</td><td>29371</td><td>birds</td><td>236415</td><td>4581108</td></tr>
<tr><td>covtype</td><td>581012</td><td>512198</td><td>covtype</td><td>2292518</td><td>34770413</td></tr>
</table>

We also compare the number of nearest-neighbor queries made by the heap and bucket based algorithms. As expected, the number of NNS queries made by the bucket-based algorithms is an order of magnitude higher than that of the heap based approach. This is due to many redundant queries performed at each round by the bucket-based algorithm. However, these queries can be performed in parallel, thus resulting in good speed-ups for this algorithm when run on many cores.

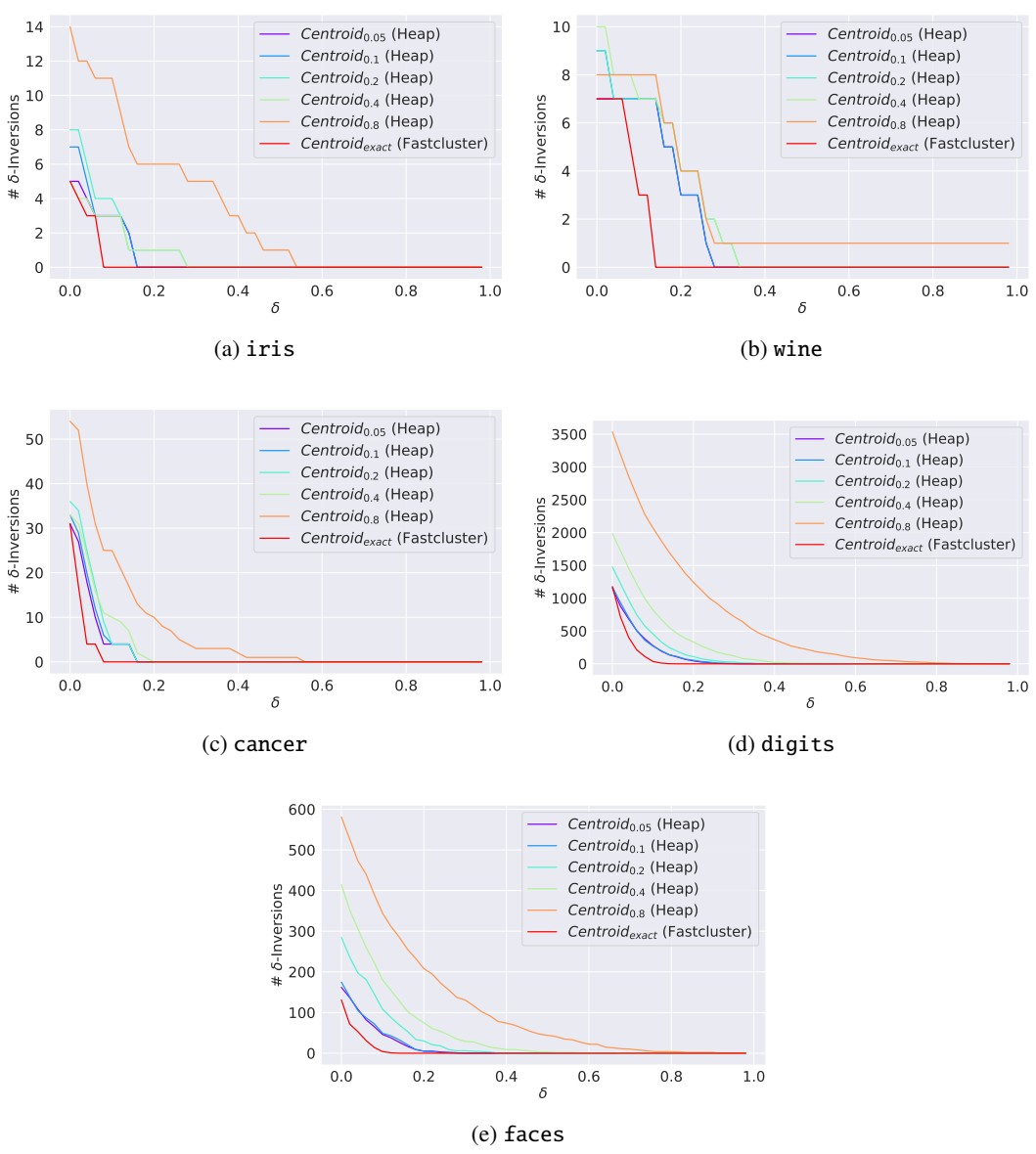

Figure 5: Non-Monotonicity of Centroid HAC: No. of $\delta$-inversions vs $\delta$

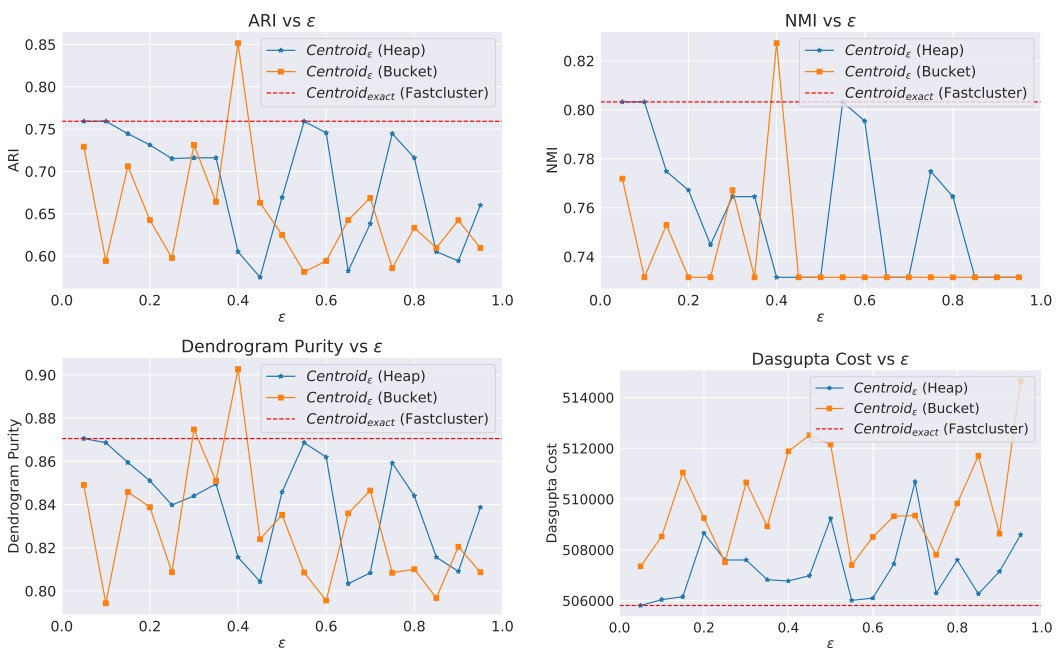

Figure 6: Quality Evaluations of the `iris` dataset

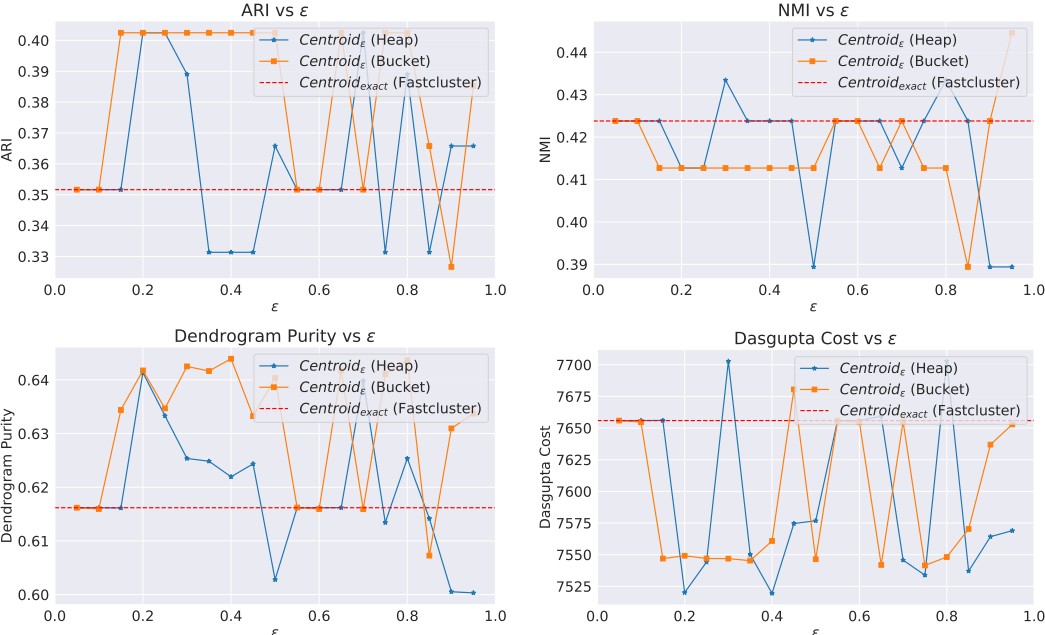

Figure 7: Quality Evaluations of the `wine` dataset

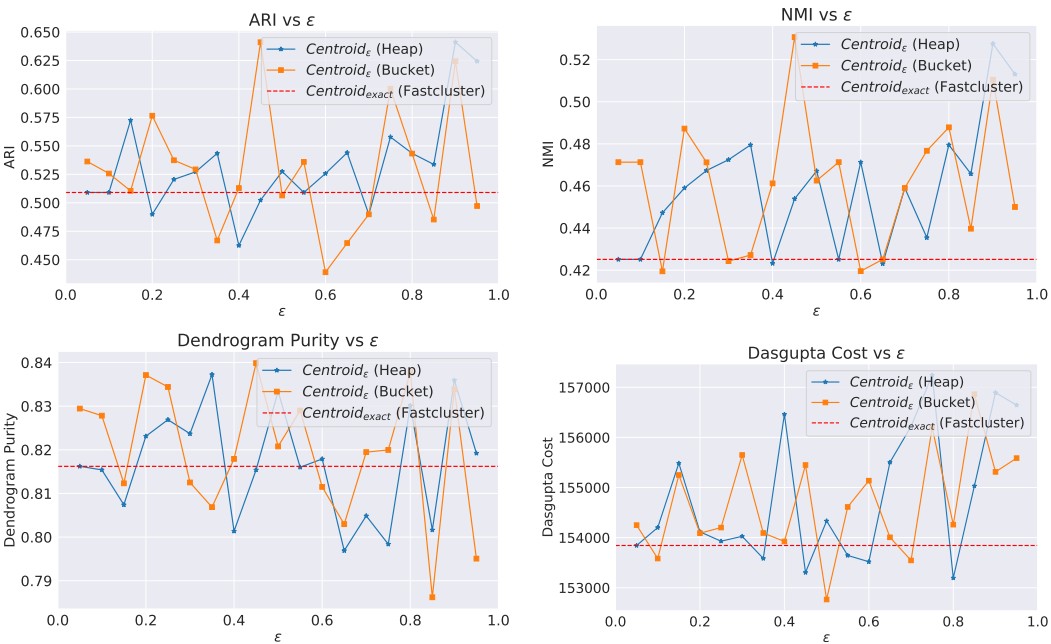

Figure 8: Quality Evaluations of the `cancer` dataset

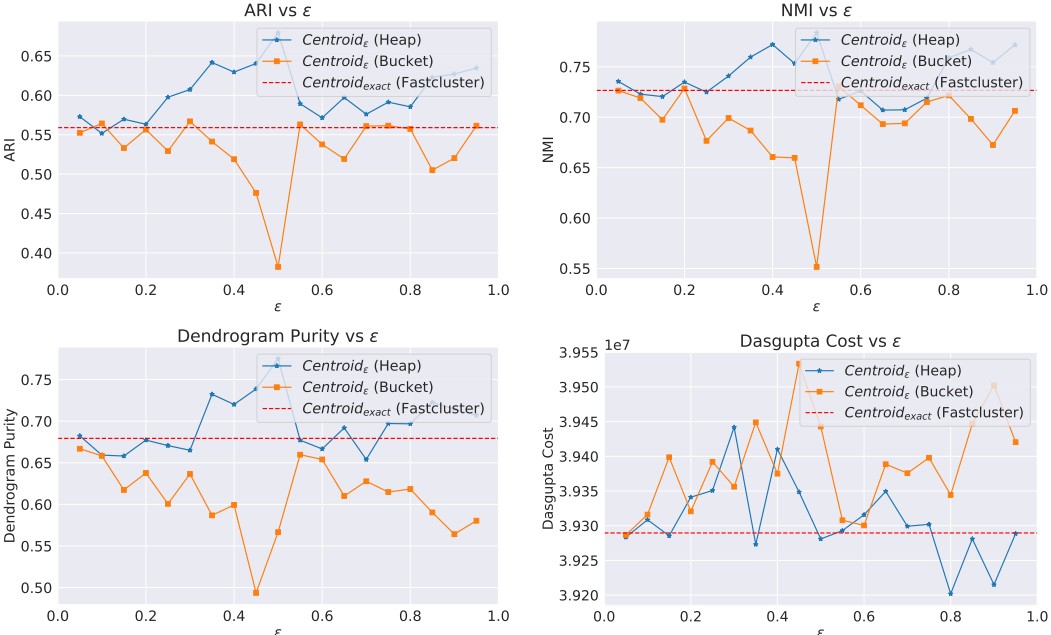

Figure 9: Quality Evaluations of the `digits` dataset

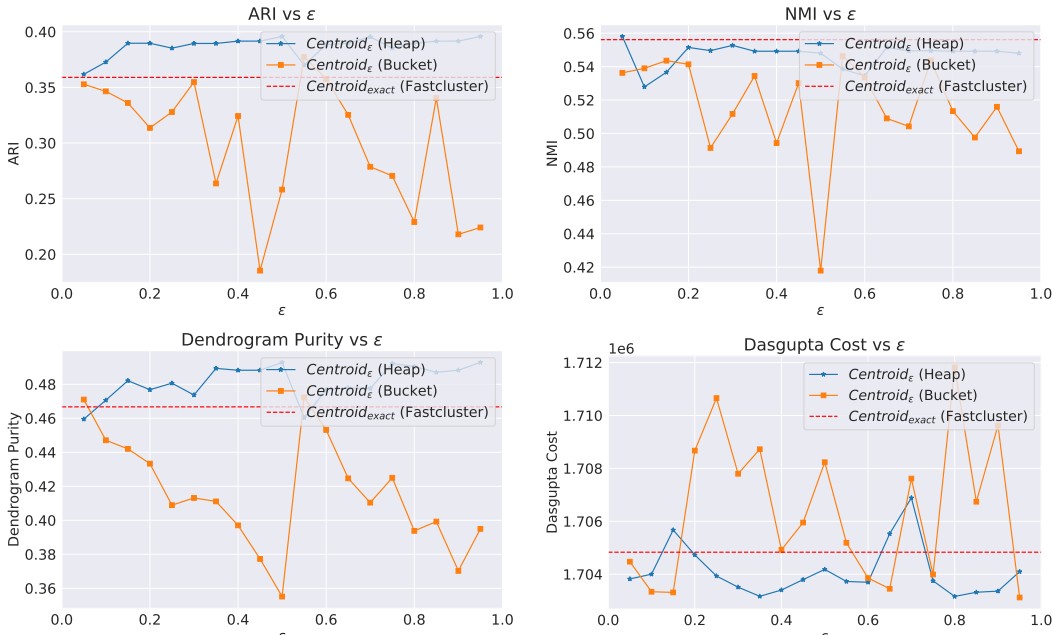

Figure 10: Quality Evaluations of the `faces` dataset

