# OpenReview forum: "Efficient Centroid-Linkage Clustering"
_NeurIPS.cc/2024/Conference — NeurIPS 2024 poster_

### Official Review · Reviewer_JGDe · 2024-06-24

**Soundness:** 4
**Presentation:** 4
**Contribution:** 4
**Rating:** 9
**Confidence:** 3

**Summary:**

The authors give an algorithm that approximates a centroid linkage clustering. Their algorithm is fast both in theory and in practice. The algorithm is based on a novel fully dynamic data structure for nearest neighbors which is of independent interest.

**Strengths:**

The proposed algorithm is a significant strengthening compared to existing algorithms, in particular from a theoretical point of view: while it is known that under some standard complexity theoretic assumptions most hierarchical clustering methods need at least (essentially) quadratic runtime, the proposed approximation algorithm significantly breaks this barrier. This is also underlined by experimental results. Interestingly, approximation does not seem to decrease the quality of the clustering. Indeed, as can be seen in Table 1, higher approximation values often give better results than "better" approximations or even optimal algorithms.

**Weaknesses:**

The techniques are very much geared towards the centroid linkage function. It could be interesting to mention how much of it could potentially generalize to other linkage functions and where the bottlenecks for full generalization lie.

**Questions:**

-Do you have any intuition as to why "worse" approximation factors often seem to give better clustering results (as your Table 1 indicates)?

**Limitations:**

The limitations are addressed in the checklist.

---

> ### Author Rebuttal · Authors · 2024-08-07
>
> We thank the reviewer for their comments and suggestions. We respond now to the specific questions of the reviewer.
>
> > The techniques are very much geared towards the centroid linkage function. It could be interesting to mention how much of it could potentially generalize to other linkage functions and where the bottlenecks for full generalization lie.
>
> This is a great question and something that we will clarify when discussing our results in the paper. We believe that our techniques should extend naturally to other linkage functions for spatial hierarchical clustering that incorporate centroids, such as Ward’s linkage. Investigating whether any centroid-based method expressed in the Lance-Williams formulation could take advantage of our approach is an interesting question for future work.
>
> On the more theoretical side, we note that our theoretical results for ANNS could be used more generally in other clustering algorithms that require adaptivity.
>
>
> > “Do you have any intuition as to why "worse" approximation factors often seem to give better clustering results (as your Table 1 indicates)?”
>
> This is an interesting question, and as briefly mentioned in the paper, we do not have any scientific explanation for this phenomenon. One possibility is that as \eps increases, the decision process slowly incorporates an approximate single-linkage-like behavior, which could be beneficial for certain datasets. We plan to include several additional results on clustering datasets with ground truth in the final version of the paper (e.g., on ImageNet, Reddit, and arXiv embeddings as discussed in our response to reviewer Dgto) which will provide more data on the effect of increasing \eps.

---

> > ### Comment · Reviewer_JGDe · 2024-08-08
> >
> > I would like to thank the authors for their rebuttal. Their answer to my second question seems plausible, and it would perhaps be interesting to compare the quality of the clusterings to the results of other methods, such as single-linkage to investigate the proposed reason.

---

> > > ### Author Response · Authors · 2024-08-14
> > >
> > > Thank you again to the reviewer for their suggestions. We will further investigate this direction by comparing with single-linkage clustering and include our findings in the revised version.

---

### Official Review · Reviewer_Dgto · 2024-07-07

**Soundness:** 3
**Presentation:** 2
**Contribution:** 3
**Rating:** 6
**Confidence:** 4

**Summary:**

The paper deals with centroid-linkage agglomerative clustering (centroid HAC), where the distance between two clusters is the distance between their centers. It presents a subquadratic time algorithm (approximate centroid HAC) that, instead of requiring the two closest clusters to be merged at each step, it allows for any two clusters to be merged if their distance is within a factor of c of that of the two closest clusters.

**Strengths:**

S1. A dynamic approximate NN search algorithm with adaptive updates is proposed.

S2. A centroid-linkage HAC algorithm is presented that exploits the approximate NN search algorithm.

S3. Theoretical performance bounds are presented and proved.

S4. Empirical evaluation indicates that the approximate centroid HAC algorithm provides clustering results very close to exact centroid HAC, but with a considerable speedup in execution time.

**Weaknesses:**

W1. It is confusing that the ANNS algorithm used in the experiments is not the same as the one proposed and analyzed in Section 3 of the paper (Algorithm 1). Since the focus of the paper is on clustering, this issue creates confusion regarding the actual contribution of this work. Perhaps the ANNS algorithm (contribution 2) is better to be presented in a forum related to data structures.

W2. It is not clear if the ‘approximate centroid HAC’ idea is novel or not (independently of the ANNS algorithm used in the implementation).

**Questions:**

Q1, Q2. See comments W1, W2 above.

Q3. How does the method compare to other approximate HAC methods in terms of NMI? In Figure 3  running time comparison is provided. In Figure 5 some results are provided on the Iris dataset, but I cannot see any general conclusions.

Q4. It is meaningless to compare using small datasets, since the optimal solution can be quickly obtained. More experiments using big would add value to the paper.

Q5. Is it possible to have a (rough) estimate of the increase in execution time as the value of \epsilon decreases?

**Limitations:**

The authors discuss the limitations. I do not see any potential negative societal impact of this work.

---

> ### Author Rebuttal · Authors · 2024-08-07
>
> We thank the reviewer for their comments and suggestions. We respond now to the specific questions of the reviewer.
>
> > Q1/W1: Coherence and clarifying why we use a different ANNS algorithm in theory and in practice.
>
> We thank the reviewer again for raising this question, which we agree is important. We present a longer discussion about this question in the overall response at the top of our rebuttal.
>
>
> > Q2/W2: Novelty of “approximate centroid HAC”
>
> We believe that the formulation of approximate HAC is certainly not novel, and there has been a number of interesting works in the past few years exploring similar notions of approximation for other linkage functions for dissimilarity and similarity (or graph) settings (see for example references [1, 5, 11, 12, 13, 21] which all deal with similar notions of approximation for other linkage functions). However, we strongly believe the techniques we introduce and leverage for obtaining our results for c–approximate HAC are novel, and have not been used in the literature before, and are not simple extensions of known techniques.
>
>
> > Q3: “How does the method compare to other approximate HAC methods in terms of NMI? In Figure 3 running time comparison is provided. In Figure 5 some results are provided on the Iris dataset, but I cannot see any general conclusions.”
>
> We thank the reviewer for this question; we will add a better explanation of how the NMI of our approximate HAC methods compares with the exact method in the main body of the paper. We included a broader discussion of these results on more datasets in the appendix, which we will move into the main body of the paper. The main result is that for modest values of \eps (e.g., \eps=0.1) the approximate method consistently gets NMI within a factor of 2% of that of the exact algorithm.
>
>
> > Q4: Using larger datasets to show more meaningful scaling results
>
> In this paper, we considered standard clustering benchmark datasets at three different scales: small, medium, and large. The small datasets (e.g., iris, faces, etc.) and medium-sized datasets (e.g., MNIST, birds, where embeddings for the latter are obtained via neural network-based methods) are labeled and are used for our quality evaluation experiments. As the reviewer correctly pointed out, the exact algorithms run very fast on the small datasets, however, they do not scale effectively to medium and larger datasets due to their quadratic running time.
>
> Notably, the approximate HAC algorithm achieves nearly 30x speedup on MNIST and 60x speedup on Birds with 192 threads; we will include these running time results in the revised version. For quality evaluation on large datasets, we plan to add three new embedding-based datasets that come with ground truth which we described in the global response. If the reviewer has any other suggestions for datasets to cluster with meaningful ground truth labels we would be delighted to include them in our paper.
>
> We would like to clarify that the benchmark datasets considered are consistent with multiple recent works on graph and pointset clustering (see for example references [13, 14, 38]). Additionally, we plan to test the scalability of our approximate HAC algorithm on larger ANN benchmark datasets (e.g., Wikipedia-Cohere ~30M) in the revised version. However, since the algorithm is sequential, we do not expect it to scale to billion-sized datasets. Developing an efficient parallel algorithm for approximate Centroid HAC and other linkage criteria is an interesting future direction to address this issue.
>
> > Q5:Execution time v/s $\epsilon$
>
> This is a great question and something we investigated—please see Figure 3(c) in the paper, which  illustrates the trend in execution time as $\epsilon$ decreases.

---

> > ### Comment · Reviewer_Dgto · 2024-08-10
> >
> > I thank the authors for the rebuttal and their efforts to respond to my comments. I am still concerned about the incoherence issue: the ANNS algorithm proposed and analyzed is different from the one used in the experiments. For a reader who wishes to apply the method, the proposed ANNS algorithm is actually irrelevant. I will increase my score from 4 to 6.

---

> > > ### Author Response · Authors · 2024-08-14
> > >
> > > We thank the reviewer for increasing their score. We will make an effort to add a detailed discussion on this gap between theory and practice in the revised version, which we hope will address their concern.

---

### Official Review · Reviewer_vHLH · 2024-07-12

**Soundness:** 3
**Presentation:** 3
**Contribution:** 3
**Rating:** 7
**Confidence:** 3

**Summary:**

This paper considers the design of approximate versions of the centroid linkage method for hierarchical clustering.  The exact version of centroid linkage requires $\Theta(n^2)$ time, but it could be possible to do better if some relaxation is allowed.  This paper shows how to get a $c$-approximate centroid linkage clustering in sub-quadratic time by using a dynamic approximate near-neighbor search data structure that has guarantees against adaptive updates.  This is non-trivial since the distance between centroids is non-monotone: after merging a pair of clusters, the resulting distance between centroids of other clusters could go down.  To complete the result, they construct such a data structure that supports insertion, deletion, and queries against adaptive updates with sub-linear running time.

An extensive experimental evaluation is performed to test a practical variant of the proposed algorithm on standard benchmark datasets against the exact centroid linkage method provided by the fastcluster library, a standard library for hierarchical clustering algorithms).  The proposed method either shows good performance or minimal loss against the exact method for a variety of clustering metrics and significant speedup over the exact method.

**Strengths:**

There has been much work on scaling up hierarchical clustering methods by considering approximation, particularly linkage methods such as single linkage, average linkage and ward's method.  In contrast, there has been less (but non-zero) work on centroid linkage.  This paper provides an elegant and cleanly described solution to this problem.

**Weaknesses:**

- The experimental evaluation isn't for the algorithm that is proposed theoretically, but instead for a variant with several practical modifications (e.g., using graph-based ANNS which may not have the same theoretical guarantees as LSH-based ones).

- For the experiments, the authors also consider running the algorithm with 192 cores.  For this case, it seems that fastcluster cannot make use of the additional cores, which seems to be an unfair comparison.  However, they do use an exact version of the new algorithm which does make use of the additional cores, providing a more fair comparison.

**Questions:**

Minor comment - the usage of $Q$ for both covering net and priority queue is somewhat confusing.  Please consider modifying this.

**Limitations:**

Limitations are made clear through a clear description of assumptions.  I don't see potential negative societal impact from this work since it develops more efficient versions of already widely-used algorithms.

---

> ### Author Rebuttal · Authors · 2024-08-07
>
> We thank the reviewer for their comments and suggestions. We respond now to the specific questions of the reviewer.
>
> > W1: The experimental evaluation isn't for the algorithm that is proposed theoretically, but instead for a variant with several practical modifications (e.g., using graph-based ANNS which may not have the same theoretical guarantees as LSH-based ones).
>
> We thank the reviewer for raising this important point, which we acknowledged in the limitations section of our submission. We present a longer discussion about this question in the overall response at the top of our rebuttal.
>
> > W2: For the experiments, the authors also consider running the algorithm with 192 cores. For this case, it seems that fastcluster cannot make use of the additional cores, which seems to be an unfair comparison. However, they do use an exact version of the new algorithm which does make use of the additional cores, providing a more fair comparison.
>
> We actually also evaluated all algorithms in the sequential setting and still obtained a speedup (even on one core) over fastcluster. The results for our single core running times are in Figure 3(a). While it is true that fastcluster can’t make use of the additional cores, we view this as an advantage of our algorithm. The results with many cores are in Figure 3(b). As the reviewer notes, the runtime of fastcluster is the same in both scenarios.
>
> > Q1: Minor comment - usage of Q for both covering net and priority queue is somewhat confusing. Please consider modifying this.
>
> Thank you for pointing this out, we will fix it.

---

> > ### Comment · Reviewer_vHLH · 2024-08-07
> >
> > Thank you for your response and addressing my questions.  My overall evaluation remains the same.

---

### Official Review · Reviewer_Pmjq · 2024-07-13

**Soundness:** 3
**Presentation:** 4
**Contribution:** 3
**Rating:** 6
**Confidence:** 4

**Summary:**

Paper studied Hierarchical clustering algorithm. HAC is popular method where n points are initially singleton clusters (leaves). At each step, two clusters are merged, and the process is continued until one giant cluster of all n points (root) remains. THe merging process gives a hierarchy or nested clusterings. Depending on which clusters are merged ,the algorithm differs.

Common methods are single linkage where the distance between clusters is in terms of min x in C_1 y in C_2 d(x,y).
Another method which is focus of current undertaking is centroid linkage, which is d(C_1, C_2) = dist(mu_1, mu_2) where mu_i is centroid vector of cluster i.
Technical point of view: Interestingly, after merging two clusters,the best inter-cluster distance can sometimes decrease, so it is not monotone increasing over time as in single linkage.

Since many lower bounds exist showing that we need theta(n^2) time to do HAC, paper studies approximate HAC, where we merge any apprxoximately closest clusters instead of exact closest clusters.

To do this, authors capture the HAC problem as that of approximate nearest neighbors over dynamic dataset. Each cluster is a point in the ANN index, and when 2 clusters merge, we capture by 2 deletions and 1 insertion. Then using a priority queue where each cluster maintains its closest distance to other clusters, the authors show how to maintain this PQ online, and use this to quickly determine which clusters to merge.

To make it all work they devise provable new algorithms for dynamic ANN algorithms using LSH which can handle adaptive inserts/deletes (which is needed for this problem). Apparently, prior work only handles non-adaptive.

**Strengths:**

* Important problem of getting sub-quadratic algorithms for hierarchical clustering, quite reasonably well written paper.
* Theoretical backing + empirical applications using the theory ideas are given, which is nice
* Good set of evaluations and the numbers are great -- quality is better and significantly faster.
* The dynamic ANN algorithm introduced in this paper ideas could be useful in other settings also.

**Weaknesses:**

Datasets chosen for evaluation can be more diverse.

The theoretical result can be presented with more rigor -- For example, I could not follow what objective function they are presenting a c-apprxoximation for the HAC problem in Theorem 5.

The technical reason/idea behind why they are able to convert a non-adaptive ANN structure to adaptive using the power-of-two size indices is not explained well, seems like the crux of the paper but missing in detail. Is there precedent for using such ideas to make non-adaptive inputs into adaptive inputs?

**Questions:**

In Line 140, don't you want the weighted average  and not the weighted sum?

What is the objective function you use to compare the quality of HAC soluton and prove approximation factor gaurantee?

Why not use more modern-day neural-network inspired clustering datasets for benchmarking, as it will be an important use-case going forward (like the OpenAi embeddings or more BERT-based embeddings).

What is SOTA known for other linkage functions in HAC like single linkage and average linkage? Do your ideas extend there also? Which linkage is most used in practice?

You cite [2, 14] for dynamic non-adaptive ANN, However I cant find the corresponding results in either. Please be more specific.

**Limitations:**

Would be good to mention a more detailed limitations section apart from those mentioned in conclusion / open problems.

---

> ### Author Rebuttal · Authors · 2024-08-07
>
> We thank the reviewer for their comments and suggestions. We respond now to the specific questions of the reviewer.
>
> > “Datasets chosen for evaluation can be more diverse.”
>
> To address this point we have assembled several new clustering datasets with ground truth from existing sources (which we will make publicly available). The datasets are described in more detail in the global response. We will include quality and scalability results on these datasets, which we hope will address the reviewers concern.
>
> > c-approximation for the HAC problem in Theorem 5
>
> Thanks for pointing this out. The definition of c-approximate HAC is given on l138. We will improve the writing to clarify this.
>
>
> > “The technical reason/idea behind why they are able to convert a non-adaptive ANN structure to adaptive using the power-of-two size indices is not explained well, seems like the crux of the paper but missing in detail. Is there precedent for using such ideas to make non-adaptive inputs into adaptive inputs?”
>
> We will add more intuition and improve the writing for the ANNS section. The power-of-two method is known as merge and reduce and is an established technique (e.g. see line 197) for turning static data structures into dynamic ones. This allows us to insert new points to query against but to support adaptive query points we introduce covering nets.
>
> Merge-and-reduce: The problem that this fixes is that ANNS data structures for Euclidean distance are static meaning we can’t insert new points to query against. The idea is that we will have buckets labeled 0, 1, … , log(n) where bucket i will hold a single static ANNS data structure with at most 2^i points. To insert a point to query against we will add it to bucket 0. Then while there is a bucket i with “loose” points that are not in an ANNS data structure (there will be at most one at any time):
> *  If there are at most 2^i total points in bucket i, build a new ANNS with all points in bucket
> *  Else move all points from bucket i to bucket i+1 as “loose” points
>
> Roughly speaking, in bucket log(n) - i it will take time (n/2^i)^(1+epsilon) to construct an ANNS but we will only ever have to construct 2^i ANNSs in that bucket. It follows that we will only spend at most n^(1+epsilon) time constructing ANNSs for each of the log(n)+1 buckets. In the actual proof we charge the running time to the points instead of the buckets. To query or delete we only have to query or delete from at most a single static ANNS in each bucket.
>
> Covering net: ANNS data structures assume that query points are non-adversarial. Suppose we are given a point p that we will want to find an approximate nearest neighbor for in some set S. If we construct the ANNS on S then whp it will return an approximate nearest neighbor for p when queried. Given a set of query points that is not too large we can union bound to say that the ANNS will return an approximate nearest neighbor for all of them.
> However, in the adaptive case we must assume that an adversary who knows the randomness of the ANNS data structure gets to pick the query point. Since we can’t union bound against the infinite possible query points we introduce covering nets. A covering net is a finite set of points such that every possible query point is near a point in the covering net. Then for every query we round it to a point in the covering net and find an approximate nearest neighbor for that point. This allows us to union bound against only the points in the covering net at the cost of picking up a small additive error in the query.
>
> > “In Line 140, don't you want the weighted average and not the weighted sum?”
>
> Yes, thank you for flagging this.
>
> > “Why not use more modern-day neural-network inspired clustering datasets for benchmarking, as it will be an important use-case going forward (like the OpenAi embeddings or more BERT-based embeddings).”
>
> We definitely agree with the importance of this use case. However, we are not aware of very many publicly available datasets of this sort that would come with ground truth clustering labels. We note that in our paper, we have used one embedding-based dataset (see Appendix E for the description of the birds dataset); we will also add three new embedding-based datasets that come with ground truth which we described in the global response. If the reviewer has any other suggestions for datasets to cluster with meaningful ground truth labels we would be delighted to include them in our paper.
>
> > “What is SOTA known for other linkage functions in HAC like single linkage and average linkage? Do your ideas extend there also? Which linkage is most used in practice?”
>
> Computing single-linkage clustering can be achieved by first computing an MST of the input points and then applying a simple post processing step. For the complexity of Euclidean MST, see https://en.wikipedia.org/wiki/Euclidean_minimum_spanning_tree#Computational_complexity.
>
> In the case of average linkage, the state of the art is a paper we cite: https://papers.nips.cc/paper_files/paper/2019/hash/d98c1545b7619bd99b817cb3169cdfde-Abstract.html
>
> However, the proposed algorithm to the best of our knowledge has not been implemented, likely due to the somewhat large overhead of efficiently computing the average linkage distance. Applying our techniques to average linkage, which seems to be a harder problem than the one we address, is an interesting open problem.
>
>
> > Would be good to mention a more detailed limitations section apart from those mentioned in conclusion / open problems.
>
> Thank you for the suggestion. We will add a section outlining two main limitations:
> *  the use of different ANNS data structure in the theoretical and empirical results.
> *  quality evaluation limited to medium-sized datasets, due to the lack of suitable publicly available embedding datasets

---

### Author Rebuttal · Authors · 2024-08-07

We thank all the reviewers for their thoughtful comments and suggestions. We reply to the questions and comments of each reviewer individually in the corresponding rebuttal fields.

# Coherence:
First, we would like to clarify a point raised by the reviewers regarding the coherence of our theoretical and experimental results, namely that the theoretical algorithm and experimental evaluation use different ANNS algorithms (a point brought up by reviewers vHLH and Dgto).

We acknowledge this limitation and would like to point out that we have been open about it in the checklist of our submitted paper. We would like to discuss the issue from two angles.

In terms of the coherence of the paper, the crux of the paper is the efficient approximate centroid HAC algorithm. In the paper we show that it gives a very efficient theoretical algorithm (using our new data structure based on LSH) and an algorithm which is very efficient in practice (using state of the art ANNS techniques). So in that sense the paper presents a complete story.

In terms of the theoretical and empirical part considering different algorithms, while ideally both settings would use exactly the same algorithm, we would like to point out that our situation is still somewhat better than in multiple other prominent clustering problems. In our case, both the theoretical and practical algorithms use the same meta-algorithm (instantiated using different data structures). On the other hand, in the case of many other clustering problems, there is an even larger gap between the theoretical and practical approaches, for example:
*  Correlation clustering - while the best known algorithms are obtained by solving a linear program (see https://dl.acm.org/doi/abs/10.1145/3618260.3649749 or https://arxiv.org/pdf/2309.17243), in practice a simple local search / Louvain-based algorithm is used (see section 4.1 of https://hal.science/hal-04251953/file/openproblems.pdf).
*  Modularity clustering - the best known approximation is obtained by solving an SDP (https://www.sciencedirect.com/science/article/pii/S0022000020301124?fr=RR-1&ref=cra_js_challenge). Again, Louvain-based algorithms are used in practice.
*  In the case of k-means, while the best known approximation is a constant obtained via the primal-dual method (https://dl.acm.org/doi/abs/10.1145/3519935.3520011), in practice Lloyd’s heuristic paired with k-means++ seeding is used which has a logarithmic approximation guarantee.
*  Balanced graph partitioning - coarsening combined with a brute-force algorithm is used to obtain the best results in practice. On the other hand, the algorithms for solving the corresponding theoretical formulations: balanced cut and multiway cut are entirely different.

# Additional Experiments:
Several reviewers also mentioned that adding more experimental results on more diverse / large datasets would be helpful. To address this, we will include quality and scaling experiments on additional labeled datasets for the revised version. First is the standard Imagenet dataset which contains around 1.2M images of everyday objects from 1000 different classes. Each image is passed through ConvNet [23] to obtain an embedding of 1024 dimensions, similar to the Birds dataset. Next, we consider two large text datasets (reddit and arxiv) from the recent MTEB work [A]. The reddit dataset consists of ~420K (embeddings of) post titles and the goal is to cluster them into (50 different) subreddits. The ArXiv dataset consists of ~730K (embeddings of) paper titles and the goal is to cluster them in (180 different) categories. The embeddings are 1024 dimensional in both arXiv and reddit datasets.

[A] Niklas Muennighoff, Nouamane Tazi, Loïc Magne, and Nils Reimers. 2022. MTEB: Massive Text Embedding Benchmark. arXiv preprint arXiv:2210.07316 (2022). https://doi.org/10.48550/ARXIV.2210.07316

---

### Decision · Program_Chairs · 2024-09-25

**Decision:**

Accept (poster)

**Comment:**

This paper gives faster algorithms for centroid linkage clustering.  There has been a general interest in speeding up notoriously unscalable hierarchical clustering methods. The paper gives and algorithm with strong theoretical and empirical evidence. The reviewers agreed these results will be of interest to the clustering community at NeurIPS.